# Termination of the unfolded protein response is guided by ER stress-induced *HAC1* mRNA nuclear retention

Laura Matabishi-Bibi[1], Drice Challal[2], Mara Barucco [3], Domenico Libri[4] & Anna Babour [1] ✉

Cellular homeostasis is maintained by surveillance mechanisms that intervene at virtually every step of gene expression. In the nucleus, the yeast chromatin remodeler Isw1 holds back maturing mRNA ribonucleoparticles to prevent their untimely export, but whether this activity operates beyond quality control of mRNA biogenesis to regulate gene expression is unknown. Here, we identify the mRNA encoding the central effector of the unfolded protein response (UPR) *HAC1*, as an Isw1 RNA target. The direct binding of Isw1 to the 3' untranslated region of *HAC1* mRNA restricts its nuclear export and is required for accurate UPR abatement. Accordingly, *ISW1* inactivation sensitizes cells to endoplasmic reticulum (ER) stress while its overexpression reduces UPR induction. Our results reveal an unsuspected mechanism, in which binding of ER-stress induced Isw1 to *HAC1* mRNA limits its nuclear export, providing a feedback loop that fine-tunes UPR attenuation to guarantee homeostatic adaptation to ER stress.

The maintenance of cellular homeostasis is ensured by stress-signaling pathways that constantly adapt gene expression programs to intra or extracellular perturbations. While failure to activate these pathways prevents stress adaptation, prolonged signaling can also culminate in cell death. Accordingly, the accumulation of unfolded proteins in the lumen of the ER, referred to as ER stress, initiates the conserved UPR that alleviates the stress[1], and sustained or unresolved ER stress is detrimental to cell viability, both in mammalian[2] and yeast cells[3,4]. In yeast the UPR is signaled by a unique ER-localized receptor-like kinase and ribonuclease, Ire1[5,6] (IRE1αβ in mammals). Activated Ire1 eliminates a 252-nucleotide translation inhibitory intron within the constitutively expressed and nuclear-exported mRNA encoding Hac1, a bZIP transcription factor[7,8]. This "RNA editing" event is completed by the tRNA ligase Rlg1 (RTCB complex in mammals) which ligates *HAC1* exons[9], thereby allowing its translation[7,10]. Hac1 is in turn imported into the nucleus, where it up-regulates expression of UPR target genes containing a conserved UPR element (UPRE) in their promoters, to relieve

ER stress[11]. Because only the edited form of Hac1 is translated, this non-canonical cytoplasmic splicing event provides the key trigger of the signaling. On the other hand, the mechanisms that regulate UPR termination are less well established. Previous studies have nonetheless provided important insights into how Ire1 activity can be attenuated, by its binding to the UPR-induced ER-resident chaperone Kar2 (BiP in mammals) which buffers its oligomerization[12], by its dephosphorylation[3,13] or by phosphoryl transfer within the Ire1 kinase[4]. However, little is known about the regulations that target *HAC1* mRNA biogenesis beyond the control of its non-canonical splicing. Quality control (QC) of gene expression is known to be initiated in the nucleus, where multiple mechanisms scrutinize early steps of gene expression[14–16]. Specifically, nuclear mRNA biogenesis and export is precisely regulated so that only accurately packaged and processed mRNA ribonucleoparticles (mRNP) are exported to the cytoplasm. This is in part achieved through nuclear retention of immature mRNPs, which we recently reported to be initiated at the chromatin level by the

[1]Univ Paris Diderot, Sorbonne Paris Cité, INSERM U944, CNRS UMR7212, Hôpital St. Louis 1, Avenue Claude Vellefaux, 75475 Paris Cedex 10, France. [2]Université Paris-Saclay, CEA, CNRS, Institute for Integrative Biology of the Cell (I2BC), 91198 Gif-sur-Yvette, France. [3]Institut Jacques Monod, Univ Paris Diderot, Sorbonne Paris Cité, CNRS, Bâtiment Buffon, 15 rue Hélène Brion, 75205 Paris Cedex 13, France. [4]Institut de Génétique Moléculaire de Montpellier, Univ Montpellier, CNRS, Montpellier, France. ✉e-mail: anna.babour@inserm.fr

nucleosome spacing-enzyme Isw1. This ATP-dependent chromatin remodeler contributes to the regular spacing and phasing of the nucleosomes over coding regions[17]. Although inactivation of *ISW1* leads to global perturbation of chromatin organization -associated with increased intragenic cryptic transcription[18] and characterized by reduced nucleosome spacing[19–21]-it has little influence on cell growth and gene expression, typified by a modest derepression of very few genes[22–25]. Strikingly however, it promotes the release from chromatin and consecutive export of nuclear-retained mRNPs[26]. While this pathway was characterized using mRNP biogenesis mutants or reporter RNAs[26], we speculated that it may operate in physiological conditions to proofread mRNA biogenesis and to regulate nuclear export of mRNA and thereby gene expression. Here we demonstrate that Isw1 nuclear RNA retention activity is crucial to fine-tune the UPR homeostatic feedback loop. By directly binding the 3'UTR of the *HAC1* transcript, Isw1 limits its nuclear export and therefore cytoplasmic splicing. Under ER stress, UPR transcriptional induction of *ISW1* fosters this nuclear mRNA retention activity, which is ultimately decisive for the accurate attenuation of this signaling pathway and for cell survival upon ER stress.

## Results

### Isw1 interacts with the 3'UTR of *HAC1*

We previously reported that Isw1 was able to directly interact with RNA, using in vivo UV cross-link followed by stringent purification of "zero-length" Isw1-RNA interactants[26]. We have now performed CRAC (crosslinking analysis of cDNA) experiments[27,28] to analyze the binding of Isw1 to its RNA targets (Fig. 1a, unpublished data). While the full genome-wide analysis of these experiments is beyond the scope of this report, we have identified that Isw1 binds the *HAC1* mRNA

(Supplementary Fig. 1a), among others (Supplementary Fig. 1b), and analyzed the corresponding reads. Inspection of these data suggested that Isw1 strongly binds to the 3'UTR of the transcript (Supplementary Fig. 1a). This was confirmed by an Isw1 RNA immunoprecipitation (RIP) assay[26] including as a control, a strain expressing a chimeric version of *HAC1*, in which its 3'UTR was replaced with that of *ACT1* (*HAC1-3'ACT1*) (Fig. 1b). Whereas the same amount of a previously reported bona fide Isw1 RNA target (i.e. *IOC2*) was co-immunoprecipitated with Isw1 in cells expressing the wild-type (WT) or the chimeric version of *HAC1*, the interaction of Isw1 with the *HAC1* transcript was significantly decreased in *HAC1-3'ACT1* cells compared to WT (Fig. 1b), supporting the prominent role of the 3'UTR region of *HAC1* in its interaction with Isw1.

### *ISW1* inactivated cells fail to terminate the UPR

The association between Isw1 and the *HAC1* mRNA prompted us to evaluate the contribution of this mRNA nuclear retention factor to the UPR. To this aim, we profiled the transcriptional UPR output in WT and *ISW1* inactivated cells, during a time-course in which ER stress was induced by a two-hour treatment with tunicamycin (Tm), a well-characterized inhibitor of N-linked glycosylation that induces the accumulation of unfolded proteins in the ER. Tm treatment was followed by a wash of the drug from the medium, thereby allowing analysis of the activation and termination phases of the UPR[3] by qRT-PCR (Fig. 2a). Monitoring *HAC1* mRNA splicing using primers specific to the spliced (**i**nduced, *HAC1i*), unspliced (*HAC1u*) and total (*HAC1Tot*) forms of *HAC1* revealed that Tm equally induced *HAC1i* formation in WT and *isw1Δ* cells (Fig. 2b, compare untreated and Tm 2H). As previously reported, the level of *HAC1i* declined gradually over time to reach its initial level four hours after Tm removal in WT cells, corresponding to

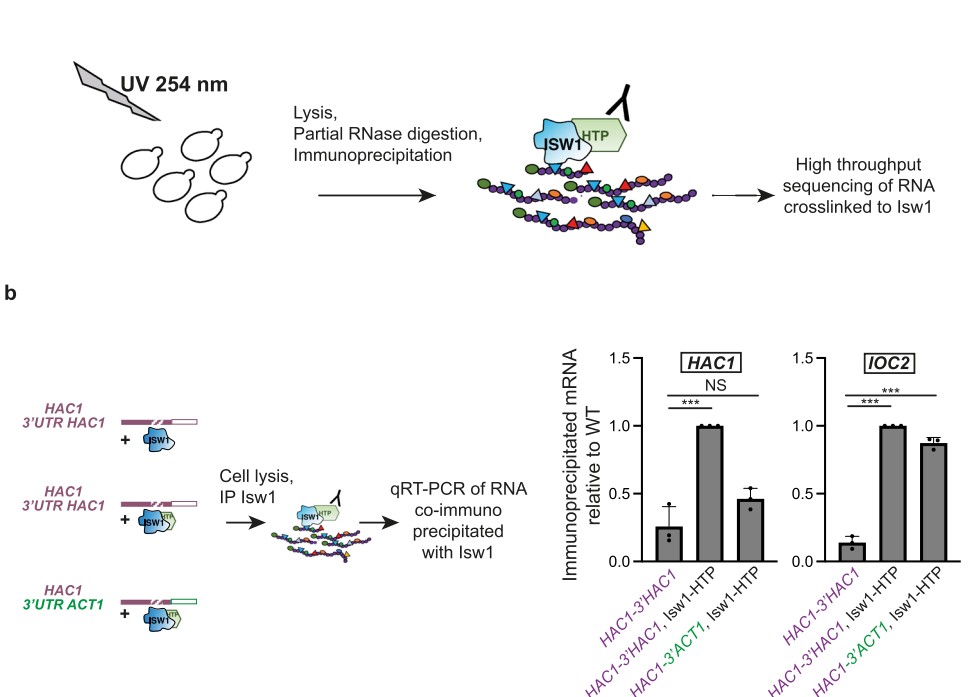

**Fig. 1 | Isw1 interacts with the 3'UTR of *HAC1* mRNA. a** Scheme depicting the identification of Isw1-bound transcripts by in vivo UV-crosslinking and analysis of cDNA. Growing yeast cells expressing dual affinity tagged (HIS6-TEV-PrA) Isw1 were UV-irradiated with 254 nm UV light to covalently crosslink RNA-protein complexes, which were immunopurified under stringent wash condition to prevent the recovery of indirect RNA targets. After ligation of linkers, crosslinked RNAs were converted to cDNA, PCR amplified, and sequenced. **b** The interaction between *HAC1* mRNA and Isw1 is mediated by its 3'UTR. RIP assays were performed with HTP

tagged Isw1 in cells expressing wild type *HAC1* (*HAC1* 3'UTR *HAC1*) or a chimeric *HAC1* (*HAC1* 3'UTR *ACT1*). Untagged WT cells were used as a negative control. The ratio of co-immunoprecipitated *HAC1* or *IOC2* RNA to total RNA relative to WT was quantified by qRT-PCR. $n = 3$ independent experiments, mean ± s.d. Unpaired one-tail *t*-test (*p* values relative to No Tag: *HAC1*, Isw1-HTP: 4.56E−04 *HAC1-3'ACT1*, Isw1-HTP: 9.95E−02 for *HAC1* and Isw1-HTP: 2.89693E−06, *HAC1-3'ACT*, Isw1-HTP: 3.55186E−05 for *IOC2*).

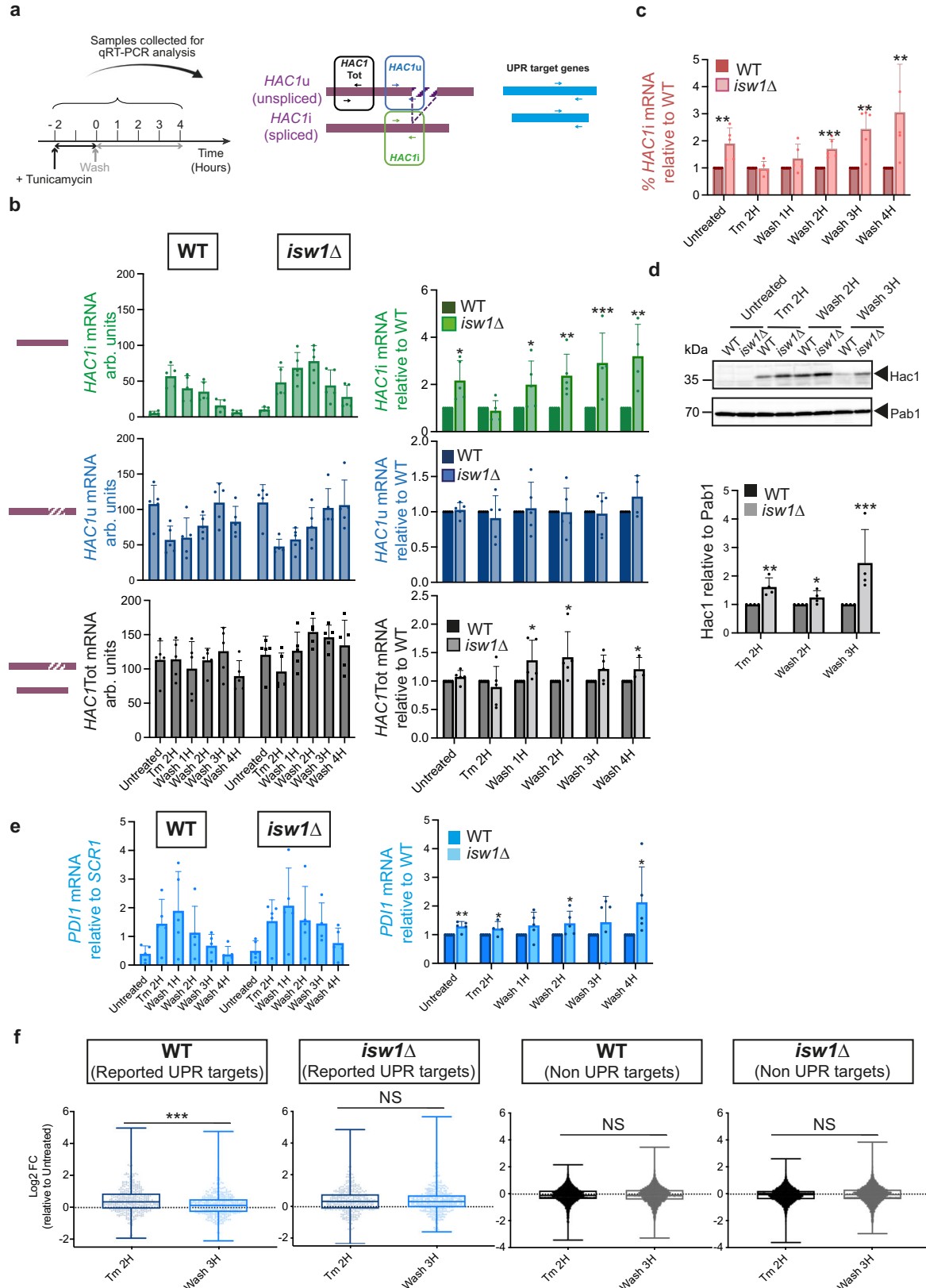

an adaptive phase characterized by an adjustment of the ER capacity that correlates with Ire1 turn-off[3,4,12]. In contrast, the decrease of *HAC1i* was delayed in *isw1Δ* cells and did not reset by four hours after Tm wash (Fig. 2b). Consistently, the percentage of spliced *HAC1* (Fig. 2c) and the expression of Hac1 protein (Fig. 2d) and Hac1 transcriptional targets (Fig. 2e, Supplementary Fig. 2a) were significantly sustained

after Tm removal in *isw1Δ* compared to WT cells. Genome-wide expression profiling of these strains across similar UPR time-courses (Untreated- Tm 2 H- Wash 3 H) reinforced these conclusions. While in WT cells the expression fold change of UPR target genes relative to untreated conditions was significantly decreased three hours after Tm wash compared to two hours Tm treatment, it remained unchanged in

**Fig. 2 | *ISW1* inactivation prevents accurate UPR termination. a** Tm UPR time course experimental setting: WT and *isw1*Δ exponentially growing cells were treated with 1 µg/mL tunicamycin for 2 h, washed and resuspended in Tm free medium. Samples were collected at $t = -2$ h (Untreated), $t = 0$ (Tm 2 H), $t = 1$ h (Wash 1 H), $t = 2$ h (Wash 2 H), $t = 3$ h (Wash 3 H), and $t = 4$ h (Wash 4 H) for analysis of *HAC1* mRNA splicing and expression of UPR target genes by qRT-PCR using the depicted primers. **b** qRT-PCR analysis of the expression levels of *HAC1*i, *HAC1*u, and *HAC1*Tot relative to *SCR1* or to WT, in WT and *isw1*Δ cells during the Tm UPR time course depicted in **b**. $n = 5$ independent experiments, mean ± s.d. **c** *HAC1* splicing is sustained in *isw1*Δ compared to WT cells during Tm UPR time course. The percentage of spliced *HAC1* (*HAC1* i / *HAC1*Tot x100) was calculated from the values obtained in **b**. $n = 5$ independent experiments, mean ± s.d. Unpaired one-tail *t*-test ($p$ values relative to WT: 4.20E−03 4.40E−01, 8.67E−02, 1.04E-03, 2.45E-03, 1.14E −02 for Untreated, Tm 2 H, Wash 1 H, Wash 2 H, Wash 3 H, Wash 4 H respectively). **d** Hac1 expression is sustained in *isw1*Δ compared to WT cells. Total protein extracts from WT and *isw1*Δ cells treated or not with 1 µg/mL Tm for 2 H were

analyzed by western blot with anti-Hac1 and Pab1 (loading) antibodies. The level of Hac1 and Pab1 was quantified. $n = 4$ independent experiments, mean ± s.d. Unpaired one-tailed *t*-test ($p$ values relative to WT: 0.004387283, 0.039668848, 0.024072787 at Tm 2 H, Wash 2 H, Wash 3 H respectively). **e** The mRNA expression level of canonical Hac1 targets is prolonged in *isw1*Δ cells compared to WT, as evaluated by qRT-PCR analysis of *PDI1* transcript expression relative to *SCR1* during Tm UPR time courses. $n = 5$ independent experiments, mean ± s.d. **f** Whisker plots of expression fold change (Log2 FC) for all UPR (defined in ref. [11]) and non-UPR target genes at Tm 2 H and Wash 3 H relative to untreated in WT and *isw1*Δ cells. Boxes extend from the 25th to 75th percentiles. The line in the middle of the box is plotted at the median. Whiskers are plotted down to the minimum and up to the maximum value, and each individual value is plotted as a point superimposed on the graph. $n = 3$ independent biological replicates. Paired two-tailed *t*-test ($p$ values for WT UPR targets <0.00001, for *isw1*Δ UPR targets 0.1408, for WT non-UPR targets 0.053, for *isw1*Δ non-UPR targets 0.1048).

*isw1*Δ cells (Fig. [2]f). Accordingly, whereas 16 out of 25 of the most robustly induced UPR genes remained differentially expressed 3 h after Tm removal in *isw1*Δ cells, only 3 of them maintained an induced state in WT cells (Supplementary Fig. 2b). Importantly, Tm similarly induced the expression of UPR target genes[11] in WT and *isw1*Δ cells (Supplementary Fig. 2c) and only the attenuation of their expression upon Tm removal was reduced in *isw1*Δ cells compared to WT. As such, differential UPR gene expression between WT and *isw1*Δ cells was only observed 3 h after Tm removal (Supplementary Fig. 2d). In addition, to exclude the possibility that the sustained UPR activation observed in *isw1*Δ results from an impaired and therefore delayed UPR activation, we conducted a careful comparison of UPR induction after a 2 h Tm treatment in WT and *isw1*Δ cells. The analysis of RNA Polymerase II (RNAPII) recruitment on 25 of the most robustly induced UPR target genes by ChIP-qPCR (Supplementary Fig. 2e), as well as the genome-wide profiling of UPR transcription using the CRAC technique to detect the position of RNAPII on UPR targets, revealed that WT and *isw1*Δ cells display comparable UPR induction profiles (Supplementary Fig. 2f, g). Finally, these results were recapitulated when UPR was triggered using DTT, a reducing agent that causes protein misfolding in the ER by counteracting disulfide bonds formation (Supplementary Fig. 2h–k). Thus, Isw1 is dispensable for *HAC1* activation but necessary for accurate UPR abatement.

### Cells lacking *ISW1* are sensitive to ER stress and defective for ER homeostasis reset

Examination of the physiological consequences of defective UPR attenuation uncovered that *ISW1* inactivation sensitized cells to ER stress (Fig. [3]a, b), a phenotype reminiscent of *ire1* mutants that activate the UPR in response to ER stress but are unable to turn-down their ribonuclease activity[3,4]. This unlikely resulted from an inability to translate UPR-induced mRNAs as no differences were observed, between WT and *isw1*Δ cells, in the steady-state level upon Tm induction of Kar2, an ER Hsp70 and canonical UPR target (Supplementary Fig. 3a). Combination of *ISW1* inactivation to an *ire1D828A* mutation that impedes UPR abatement by mimicking an active RNase conformation[3] led to increased Tm sensitivity of the double mutant compared to each single mutant (Fig. [3]b, c), implying that Isw1 and Ire1 intervene in two independent pathways of UPR termination. This stress sensitivity phenotype was not general, as no difference in sensitivity to oxidative, osmotic or mitochondrial stress was observed between WT and *isw1*Δ cells (Fig. [3]d, Supplementary Fig. 3b). To gauge how sustained UPR activation influences the ER folding capacity, we monitored, during Tm-time-courses, the glycosylation status of Protein disulfide isomerase (Pdi1), an ER-resident protein that supports protein folding by catalyzing disulfides. Pdi1 features N-glycosylation at five sites[29], which can be abolished upon deglycosylation with PNGase, resulting in increased mobility of the protein by SDS PAGE

(Supplementary Fig. 3c). When glycosylation was inhibited for two hours with tunicamycin, Pdi1 was detected as two bands, corresponding to the glycosylated and unglycosylated forms of the protein, in both WT and *ire1*Δ cells. In WT cells, upon removal of the drug and during the abatement phase of the UPR, Pdi1 progressively recovered its initial glycosylation status. In *ire1*Δ cells, Pdi1 remained deglycosylated, reflecting the inability of cells that are unable to initiate the UPR to alleviate the stress (Supplementary Fig. 3d). Strikingly, upon identical experimental settings, the kinetics of Pdi1 glycosylation was delayed in *isw1*Δ and *ire1D828A* cells compared to WT (Fig. [3]e), an effect that was strengthened in *isw1*Δ *ire1D828A* double mutant (Fig. [3]f). This delay is indicative of an altered restoration of the ER oxidative protein folding capacity. Thus, *isw1*Δ and *ire1D828A* cells present improper UPR termination and increased sensitivity to ER stress accompanied with defective reestablishment of ER homeostasis.

### Expression of *HAC1* splicing machinery, stability of *HAC1* mRNA and kinetics of Ire1 cluster assembly and disassembly are unaffected by *ISW1* inactivation

We next investigated the molecular mechanism underlying the persistence of *HAC1* splicing during the termination phase of the UPR. Transcriptional analysis indicated that the expression level of *HAC1* mRNA splicing machinery (*IRE1*, *RLG1*) was unaffected by *ISW1* inactivation (Supplementary Fig. 4a). We then asked whether sustained *HAC1* splicing could result from a stabilization of the *HAC1* transcript consequent to *ISW1* inactivation. Monitoring the stability of both forms of *HAC1* mRNA following transcriptional inhibition in cells previously treated (*HAC1*i) or not (*HAC1*u) with DTT to allow for *HAC1*i production (Supplementary Fig. 4b) revealed that ER stress stabilized *HAC1*u, in agreement with a previous report[30] (Supplementary Fig. 4c). However, *ISW1* inactivation did not affect the stability of either forms of *HAC1*. Finally, we considered that the impaired UPR attenuation observed in *isw1*Δ cells could be due to sustained Ire1 activation. To evaluate the kinetics of Ire1 activation and deactivation, we examined the kinetics of Ire1 clusterization and declusterization into foci, to which *HAC1* mRNA is recruited upon UPR induction for optimal splicing and which disappear upon UPR abatement[31–36]. *ISW1* inactivation did not alter the kinetics of Ire1 foci formation or dissociation (Supplementary Fig. 4d) examined in strains expressing at similar levels (Supplementary Fig. 4e) a GFP-tagged version of Ire1 that preserves its functions[31]. We concluded that the UPR termination defect detected in *isw1*Δ cells does neither result from an increased expression of the "UPR splicing machinery" nor from a stabilization of the *HAC1* transcript nor defective Ire1 deactivation.

### Isw1 is required for *HAC1* mRNA nuclear retention

Since Isw1 binds and retains transcripts in the nucleus[26], we reasoned that its inactivation may influence the kinetics of *HAC1* mRNA nuclear

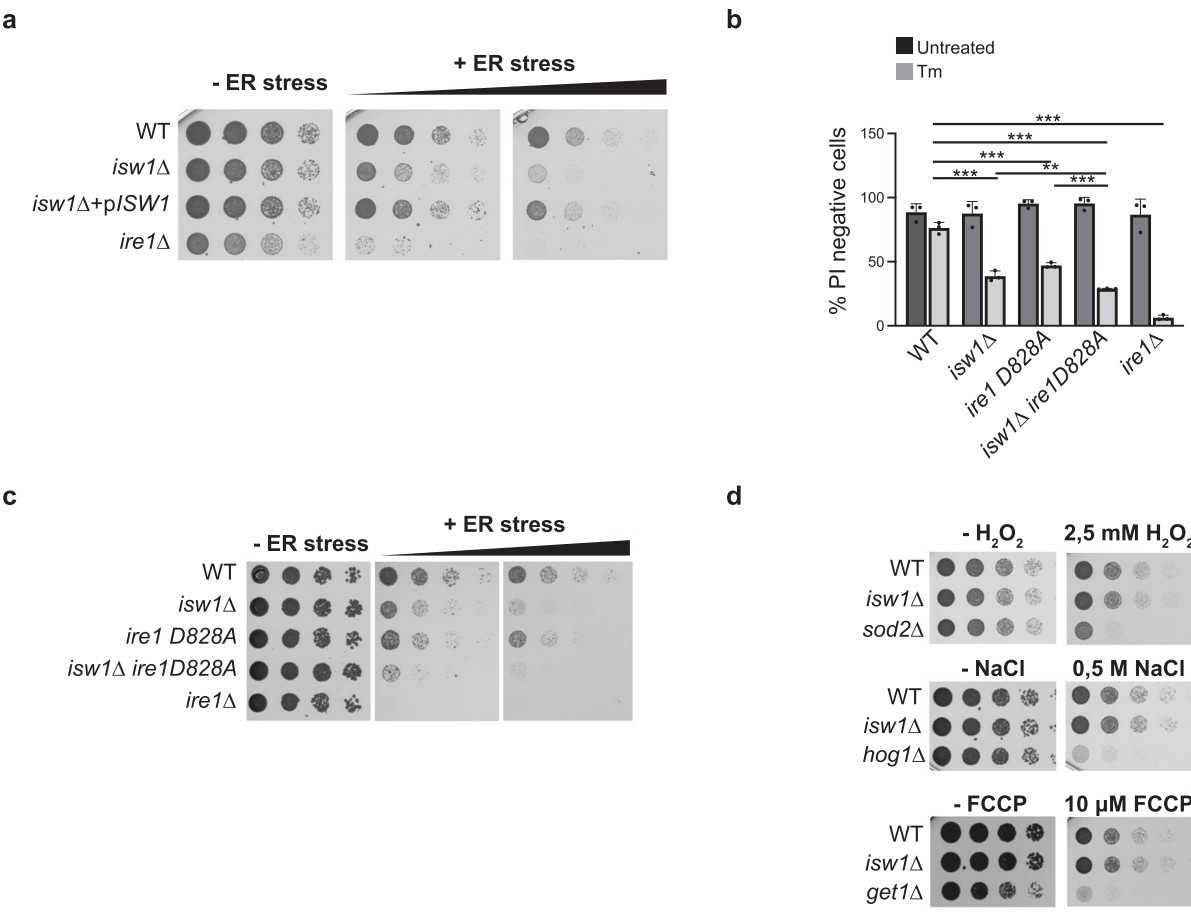

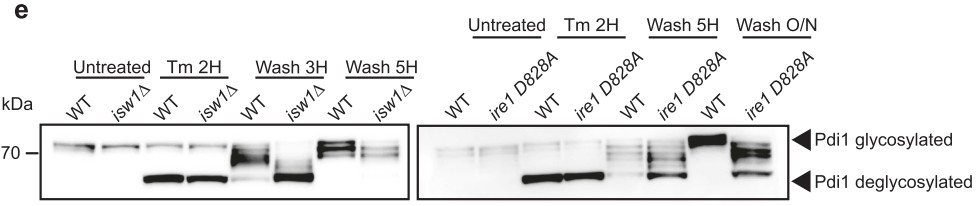

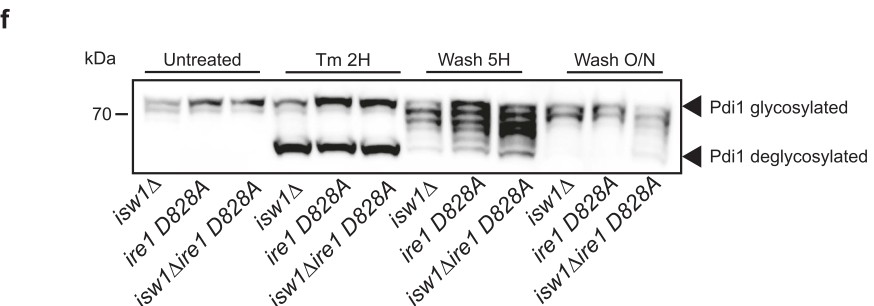

export. We therefore implemented "transcriptional chase" experiments in which the expression of *HAC1* was turned off by the addition of the transcriptional inhibitor phenanthroline to the growth media. This allows to follow the fate of transcripts synthesized before transcription inhibition (t0, Fig. 4a), by RNA Fluorescence In Situ Hybridization (FISH) using *HAC1* exons-specific probes (Supplementary

Fig. 5a). As previously reported, the distribution of the *HAC1* transcript was mainly cytoplasmic[37], with a nuclear dot corresponding to its transcription site detected upon normal conditions. In WT cells, the percentage of cells displaying an accumulation of transcripts in this nuclear dot decreased during the course of the transcriptional inhibition, reflecting the export of the corresponding transcripts.

**Fig. 3 | *ISW1* inactivated cells are sensitive to ER stress and defective for ER homeostasis reset. a** *isw1Δ* cells are more sensitive than WT to tunicamycin: fivefold serial dilutions of the indicated strains grown for 3 days at 30 °C on selective media containing (+ER stress) or not (−ER stress) 0.4 µg/mL or 0.6 µg/ mL Tm. **b** Viability assay. Exponentially growing cells were treated or not with 1 µg/mL Tm for 24 H and stained with propidium iodide (PI). The percentage of PI negative cells is plotted. A minimum of 300 cells per condition were counted. $n = 3$ independent experiments, mean ± s.d. Unpaired one-tail $t$-test ($p$ values relative to WT Untreated: 0.445937983, 0.093299173, 0.110093567, 0.41523827 for *isw1Δ*, *ire1D828A*, *isw1Δ ire1D828A* and *ire1Δ* respectively. $p$ values relative to WT Tm: 0.000201098, 0.000243914, 2.46522E-05, 7.60488E-06 for *isw1Δ*, *ire1D828A*, *isw1Δ ire1D828A* and *ire1Δ* respectively. $p$ value for *ire1D828A isw1Δ* relative to *isw1Δ* 0.00662224. $p$ value for *ire1D828A isw1Δ* relative to *ire1D828A*

5.76554E-05. **c** *isw1Δ* and *ire1D828A* mutations have compounding effects. Five-fold serial dilutions of the indicated strains grown for 3 days at 30 °C on selective media containing (+ER stress) or not (−ER stress) 0.4 µg/mL or 0.6 µg/mL Tm. **d** *isw1Δ* cells are not more sensitive than WT cells to oxidative, osmotic and mitochondrial stresses. Fivefold serial dilutions of the indicated strains grown for 2 days at 30 °C with or without 2.5 mM $H_2O_2$, 0.5 M NaCl or 10 µM Carbonyl cyanide-*p*-trifluoromethoxyphenylhydrazone (FCCP), a potent uncoupler of mitochondrial oxidative phosphorylation. *sod2Δ*, *hog1Δ* and *get1Δ* mutants respectively serve as positive controls for the effect of the stresses. **e, f** Pdi1 re-glycosylation during Tm UPR time course. Total protein extracts from WT, *isw1Δ*, *ire1D828A*, and *isw1Δ ire1D828A* cells collected during Tm UPR time courses were analyzed by western blot with anti-Pdi1 antibodies. Three independent experiments were performed with similar results.

Strikingly, this decrease was faster in *isw1Δ* than in WT cells (Fig. 4b), indicating that *ISW1* inactivation fosters the release of nuclear retained *HAC1* transcript from chromatin. To further strengthen this observation, we quantified, the intensities of the *HAC1* mRNA signal in the nucleus and the cytoplasm of untreated cells, an approach that we validated on a manually quantified and previously published data set[26] (Supplementary Fig. 5b) as well as by evaluating the number of cytoplasmic *HAC1* transcripts through manual counting (Supplementary Fig. 5c). This analysis uncovered a significant decrease in *HAC1* nuclear signal intensity, mirrored by an increase in the cytoplasmic *HAC1* mRNA signal, recapitulated by a significant decrease in the ratio of nuclear to cytoplasmic *HAC1* mRNA signal intensity in *isw1Δ* compared to WT cells (Fig. 4c). This indicates that the absence of Isw1 correlates with a reduction of *HAC1* mRNA steady-state nuclear localization. Similar results were observed when *HAC1* mRNA localization was monitored after UPR induction or during UPR abatement (Supplementary Fig. 5c, d). In a complementary approach, we used the measurement of transcripts co-immunoprecipitated with Cbp20, a subunit of the nuclear cap-binding protein, as a proxy for nuclear mRNA quantification. The amount of *HAC1* transcripts co-immunoprecipitated with Cbp20 was significantly decreased in *isw1Δ* compared to WT cells, further supporting that inactivation of *ISW1* reduces the steady-state nuclear localization of the *HAC1* transcript. A similar effect was observed for *IOC2*, a previously defined Isw1 target, consistent with the reported nuclear mRNA retention activity of Isw1[26] (Fig. 4d). Altogether, our results demonstrated that Isw1 is required for the nuclear retention of *HAC1* mRNA.

### *HAC1* IBM1 deletion phenocopies *ISW1* inactivation

To evaluate the influence of the Isw1-*HAC1* mRNA interaction on the UPR, we sought to design a mutant version of *HAC1* impaired for its ability to bind Isw1. The *HAC1-3'ACT1* chimera (Fig. 1b) could not be utilized for this purpose, due to its lack of a *HAC1*'s 3'UTR-located bipartite element (3'BE) which targets the transcript to Ire1 clusters for effective splicing[31]. We, therefore, determined the positions of Isw1 binding to the *HAC1* transcript thanks to the CRAC data that had identified *HAC1* as an Isw1 target transcript. These binding sites were clustered in two main regions within the 3'UTR of the *HAC1* mRNA that we named IBM (Isw1 Binding Motif) 1 and 2 (Fig. 5a). We generated two mutants, *hac1ΔIBM1* and *hac1ΔIBM2*, deleted for the two Isw1 binding motifs located from either side of *HAC1*'s 3'BE (Fig. 5a). Both mutants were more sensitive to Tm than WT cells (Fig. 5b, c Supplementary Fig. 6a). However, *hac1ΔIBM2* cells presented defective UPR activation (Supplementary Fig. 6b), in line with their hypersensitivity to ER stress. This suggested that IBM2 deletion may perturb the structure of the 3'BE-containing stem and thus precluded further exploitation of this mutant. In contrast, *HAC1*i was induced at similar levels in WT and *hac1ΔIBM1* cells (Supplementary Fig. 6c compares WT and *hac1ΔIBM1*), indicating that IBM1 deletion did not significantly perturb the structure of the 3'BE. Isw1 RIP confirmed that deletion of *HAC1* IBM1 significantly reduces the amount of *HAC1* transcript co-immunoprecipitated with

Isw1 (Fig. 5d). Remarkably, deletion of IBM1 led to defective UPR attenuation (Fig. 5e and Supplementary Fig. 6c raw data), decreased steady-state *HAC1* mRNA nuclear localization (Fig. 5f), phenocopying *ISW1* loss-of-function. In addition, IBM1 deletion was epistatic to inactivation of *ISW1* (Fig. 5b, c, e, f, Supplementary Fig. 6c), indicating that both mutations affect the same pathway. Finally, comparison of the UPR transcriptional output, the Tm sensitivity, and the subcellular localization of the *HAC1* transcript in WT, *isw1Δ*, *hac1Δ3'BE* (Δ3'BE) and *isw1Δ*, *hac1Δ3'BE* cells reinforced the distinction between the 3'BE and the IBM regions. As previously reported, Δ3'BE cells were defective for *HAC1* mRNA splicing and sensitive to ER stress[31] (Supplementary Fig. 6d, e). They showed delayed *HAC1*i production, presumably resulting from a compensatory increased expression of total *HAC1* (Supplementary Fig. 6d) and unperturbed *HAC1* mRNA subcellular localization (Supplementary Fig. 6f). Co-inactivation of *ISW1* and the 3'BE resulted in increased levels of *HAC1*Tot and prolonged *HAC1*i expression, reduced *HAC1* mRNA nuclear localization compared to WT and increased Tm sensitivity of the double mutant compared to each single mutant (Supplementary Fig. 6d–f), implying that both mutations act in different processes. Taken together, these results indicate that the phenotype of *isw1Δ* cells is recapitulated by deletion of the *HAC1* 3'UTR-located IBM1 sequence, which is distinct from the 3'BE, and that mediates the interaction of the *HAC1* transcript with Isw1.

### Overexpression of Isw1 negatively regulates the UPR

Since Isw1 can restrict cytoplasmic localization of transcripts, we wondered whether its overexpression could rescue mutants defective for UPR termination, by increasing *HAC1* mRNA nuclear localization and thereby limiting its splicing. We therefore overexpressed Isw1 in WT or *ire1D828A* cells (Supplementary Fig. 7a), which led to an increased steady-state nuclear localization of the *HAC1* transcript (Supplementary Fig. 7b). As reported, the growth and viability of *ire1D828A* cells upon Tm treatment was severely affected compared to WT (Fig. 6a, b), which was associated with decreased but sustained *HAC1* splicing upon 2-hour Tm induction and after wash-out of the drug respectively (Supplementary Fig. 7c, d) indicating defective UPR termination. Remarkably, overexpression of Isw1 partially rescued the growth of *ire1D828A* cells in the presence of Tm, while it was slightly deleterious to the growth of WT cells (Fig. 6a, b). Monitoring the transcriptional UPR output in these cells during Tm-induced UPR time-courses revealed that Isw1 overexpression correlated with a significant reduction of *HAC1* splicing in WT—in line with a significant reduction of Hac1 expression (Supplementary Fig. 7e)—and *ire1D828A* cells (Fig. 6c, d). In addition, in *ire1D828A* cells, this was associated with an improvement of the recovery of Pdi1's glycosylation (Fig. 6e lanes 11, 12), suggesting a recovery of the ER oxidative protein folding capacity. In contrast, Isw1 overexpression marginally delayed Pdi1's glycosylation recovery in WT cells (Fig. 6d, lanes 9, 10), which may result from the decrease in *HAC1* mRNA splicing. Thus, Isw1 overexpression is able to influence the UPR, likely by limiting *HAC1* splicing.

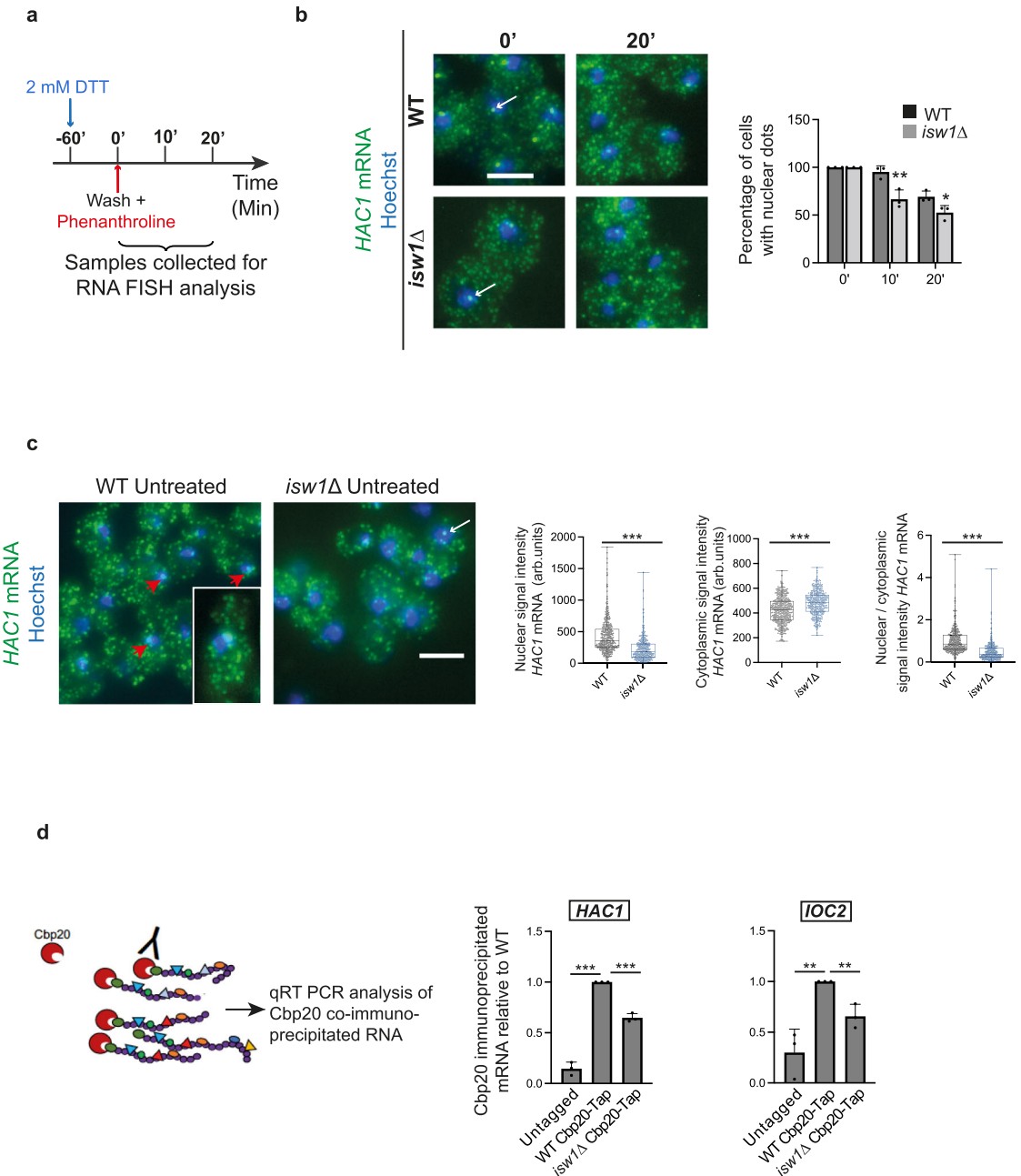

**Fig. 4 | Isw1 is required for *HAC1* mRNA nuclear retention. a** Transcription shut-off experimental setting: Cells were treated with 2 mM DTT for 2 H. At *t* = 0, cells were washed and transcription was blocked by addition of 0.2 mg/mL phenanthroline in fresh media. Samples were collected for analysis at *t* = 0, 10′ and 20′. **b** The subcellular localization of the *HAC1* transcript after blocking transcription with phenanthroline was analyzed in WT and *isw1Δ* cells by FISH using Quasar®670 -*HAC1* probes. White arrows point to transcription sites. Scale bar, 5 µm. For each time point, the percentage of cells showing a nuclear dot was scored. *n* = 3 independent experiments, mean ± s.d. Unpaired one-tail *t*-test (*p* values relative to WT: 0.006583954 and 0.0194876 for10′ and 20′ respectively). **c** *HAC1* mRNA subcellular localization was analyzed by FISH using Quasar®670-*HAC1* probes. White arrows point to transcription sites. Red arrows highlight nuclei with high- intensity *HAC1* mRNA signal (see also inset). Scale bar, 5 µm. Nuclear and cytoplasmic Quasar®670-*HAC1* signal intensities were measured in at least 200 cells. Unpaired two-tailed *t*-

test (*p* values < 0.0001 for nuclear, cytoplasmic and nuclear/cytoplasmic signal intensities). Signal intensities were plotted into whiskers plots. Boxes extend from the 25th to 75th percentiles. The line in the middle of the box is plotted at the median. Whiskers are plotted down to the minimum and up to the maximum value, and each individual value is plotted as a point superimposed on the graph. Two independent biological replicates were performed with similar results. **d** The Cbp20-*HAC1/IOC2* mRNA interactions are reduced in *isw1Δ* cells. RNA immuno-precipitation experiments (RIP) were performed with TAP-tagged Cbp20 WT and *isw1Δ* cells. The ratio of co-immunoprecipitated *HAC1* or *IOC2* (Isw1 target) RNA to total RNA present in each strain relative to WT was quantified by qRT-PCR. Untagged WT cells were used as a negative control. *n* = 3 independent experiments, mean ± s.d. Unpaired one-tail *t*-test (*p* values relative to WT Cpb20-Tap: Untagged: 1,0982E-05, *isw1Δ* Cbp20-Tap: 6,55914E-05 for *HAC1*; *p* values relative to WT Cpb20-Tap: Untagged: 0,003149472, *isw1Δ* Cbp20-Tap: 0,003733708 for *IOC2*).

## *ISW1* is itself induced by the UPR

*ISW1* was previously described in a genome-wide analysis as a UPR transcriptional target[11], suggesting that UPR-mediated *ISW1* induction could participate in a negative loop of UPR regulation. We verified the

induction of *ISW1* by ER stress (Fig. 7a, b) and analyzed the effect of stress on the interaction of Isw1 with RNA using the RIP assay. ER stress increased the amount of mRNA co-immunoprecipitated with Isw1 (Fig. 7c), supporting the possibility that this enhanced Isw1-*HAC1*

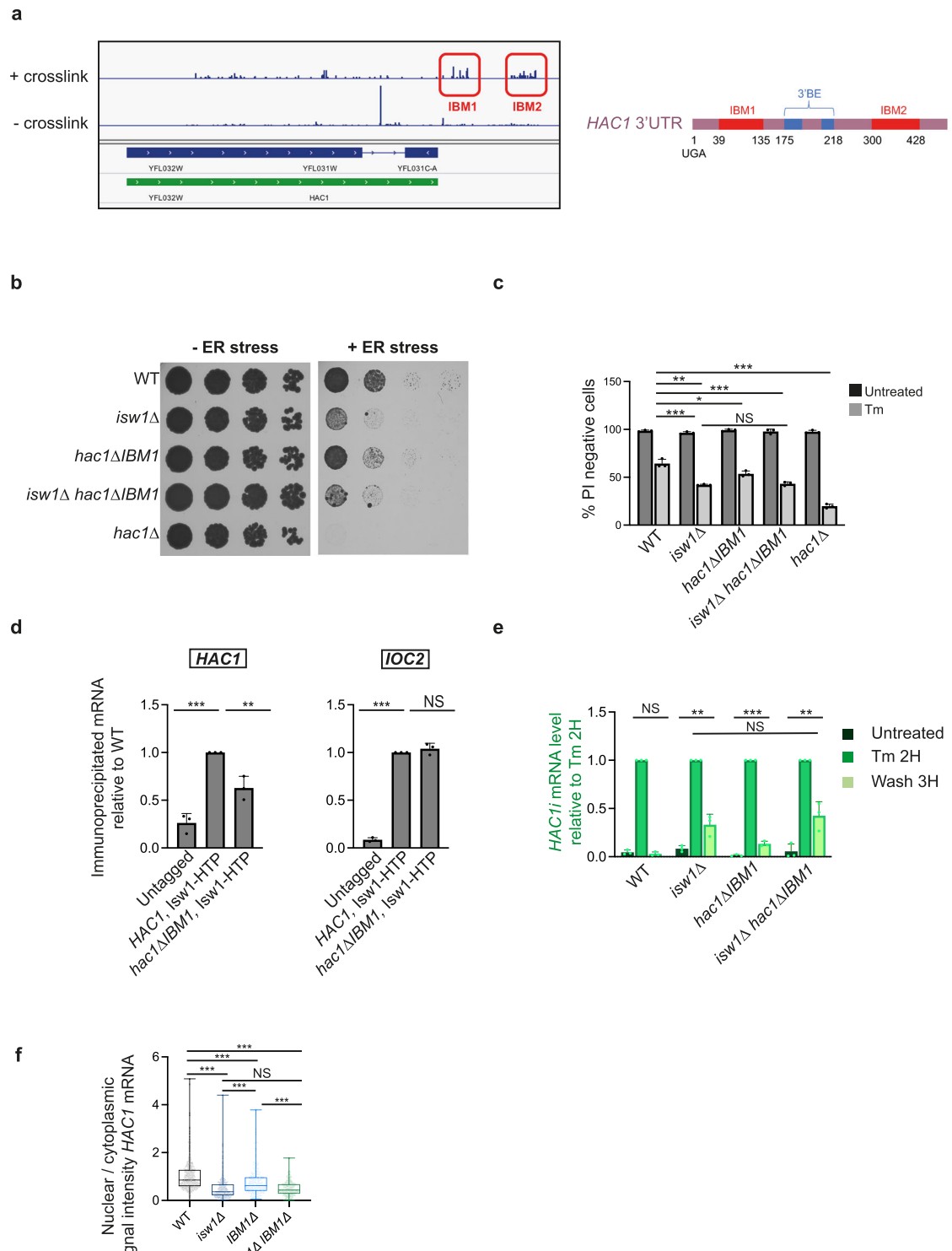

mRNA interaction could contribute to the attenuation phase of the UPR by limiting *HAC1* mRNA nuclear export. Swapping of the *ISW1* promoter region for that of *IOC4*, a non-UPR target gene with a similar transcription rate[38], prevented its ER stress induction (Fig. 7a, b). Accordingly, cells unable to induce *ISW1* in response to ER stress showed defective UPR attenuation as evaluated by monitoring *HAC1i* level during Tm-induced UPR time-course (Fig. 7d, Supplementary Fig. 8a, unnormalized) and a slight but consistent increase in sensitivity to tunicamycin (Fig. 7e, f). Ultimately, our results support a model in which Isw1 stands as a novel actor of the UPR. ER stress-induced Isw1, through its ability to directly bind the 3'UTR of the *HAC1* transcript,

limits its nuclear export and therefore cytoplasmic splicing, allowing for a negative feed-back tuning of the UPR, necessary for its accurate termination (Fig. 7g).

## Discussion

We previously reported that the chromatin remodeler Isw1, although deprived of canonical RNA binding domains, "moonlights" as an RNA binding protein (RBP) that prevents untimely nuclear export of maturing or improper mRNAs. In the present study, we underscore a case of molecular cross-talk between this newly evidenced mRNA quality control pathway and the evolutionarily conserved UPR whose

**Fig. 5 | Isw1 interacts with *HAC1* mRNA 3′UTR. a** Snapshot showing the positions of the deletions induced upon Isw1 cross-linking to *HAC1* transcript. Isw1 CRAC experiments allow determining the positions to which a protein is crosslinked to an RNA target, owing to the errors generated by the reverse transcription reaction, that lead to deletions or mutations in the sequence reads, precisely at the sites of crosslink[27,68]. Red boxes indicate two major deletion sites in the 3′UTR of the *HAC1* transcript named IBM1 and IBM2 (for Isw1 Binding Motif). **b** *ISW1* and IBM1 deletions sensitize cells to Tm without showing additive effects. Fivefold serial dilutions of the indicated strains grown for 3 days at 30 °C with (+ER stress) or without (−ER stress) 0.6 µg/mL Tm. **c** Viability of the indicated strains was analyzed as described in Fig. 3b. *n* = 3 independent experiments, mean ± s.d. Unpaired one-tail *t*-tests (*p* values relative to WT Untreated: 0.188927938, 0.188927938, 0.186383871, 0.190815197 for *isw1Δ*, *hac1Δ*IBM1, *isw1Δhac1Δ*IBM1 and *hac1Δ* respectively. *p* values relative to WT Tm: 5.19E−04, 1.29E−02, 8.60E−04 and 5.00227E−05 for *isw1Δ*, *hac1Δ*IBM1, *isw1Δhac1Δ*IBM1 and *hac1Δ* respectively. *p* value for *hac1Δ*IBM1 *isw1Δ* relative to *isw1Δ* 1.99E−01. *p* value for *hac1Δ*IBM1 *isw1Δ* relative to *hac1Δ*IBM1 3.60E−03). **d** Deletion of *HAC1* IBM1 domain reduces the amount of *HAC1* transcript co-immunoprecipitated with Isw1. RIP experiments were performed in HTP tagged Isw1 cells expressing wild type *HAC1* or *hac1Δ IBM1*. Untagged WT cells were used as a negative control. The ratio of co-immunoprecipitated RNA to total relative to WT was quantified by qRT-PCR. *n* = 3 independent experiments, mean ± s.d. Unpaired one-tail *t*-tests (*p* values relative to WT (*HAC1*, Isw1-HTP): 1.01E−04 and 3.14E−03 for Untagged and *hac1Δ*IBM1, Isw1-HTP respectively for *HAC1* target. *p* values relative to WT (*HAC1*, Isw1-HTP): 1.06396E−07 and 0.1510 for Untagged and *hac1Δ*IBM1, Isw1-HTP respectively for *IOC2* target). **e** *hac1Δ IBM1* cells are defective for UPR termination. *HAC1i* relative to *SCR1* mRNA expression during Tm UPR time courses, normalized to Tm 2H, as evaluated by qRT-PCR in the indicated strains. *n* = 4 independent experiments, mean ± s.d. Unpaired one-tail *t*-tests (*p* values Wash 3H relative to Untreated: 3.67E−01, 7.25E−04, 4.96E−02 and 6.40E−04 for WT, *isw1Δ*, *hac1Δ*IBM1 and *isw1Δhac1Δ*IBM1 strains respectively. *p* value Wash 3H *isw1Δhac1Δ*IBM1 relative to *isw1Δ*: 1.15E−01). Unnormalized data are shown in Supplementary Fig. 5c. **f** Deletion of IBM1 reduces the steady-state nuclear localization of *HAC1* mRNA. *HAC1* mRNA subcellular localization was analyzed by FISH using Quasar®670-*HAC1* probes in the indicated strains. The nuclear to cytoplasmic ratio of *HAC1* mRNA signal intensity was quantified. As in Fig. 4c, Nuclear and cytoplasmic Quasar®670-*HAC1* signal intensities were measured in at least 200 cells. Unpaired two-tailed *t*-test (*p* values < 0.0001 for *isw1Δ*, *hac1Δ*IBM1 and *isw1Δhac1Δ*IBM1 strains relative to WT and for *hac1Δ*IBM1 relative to *isw1Δ* and to *isw1Δhac1Δ*IBM1. *p* value 0.6402 for *isw1Δhac1Δ*IBM 1relative to *isw1Δ*). Nuclear to cytoplasmic ratio were plotted into whiskers plots. Boxes extend from the 25th to 75th percentiles. The line in the middle of the box is plotted at the median. Whiskers are plotted down to the minimum and up to the maximum value, and each individual value is plotted as a point superimposed on the graph. Two independent experiments were performed with similar results.

---

activation strictly relies on the cytoplasmic splicing of the Hac1 encoding mRNA. We show that the direct binding of Isw1 to the 3′UTR of *HAC1* mRNA allows precise controlling of its nuclear export, and therefore of its translation and of UPR signaling. *ISW1* being itself an UPR target gene, we propose that ER stress-induced Isw1 is a key effector of the negative feedback loop that abates UPR signaling.

Our CRAC experiment demonstrates, as previously reported[26], the binding of Isw1 to multiple mRNA species whose nuclear export are expected to be limited by Isw1. While we cannot exclude that these putative Isw1 targets might also impact UPR signaling, we assume that Isw1 binding to *HAC1* mRNA has stronger physiological consequences because it not only regulates the level of the transcript in the cytoplasm but also affects its splicing and thereby the synthesis of the short-lived Hac1 protein.

Although the effect of *ISW1* inactivation on UPR signaling may seem modest, it stands in a similar range to that of previous reports. As such, inactivation of PDIA6, a protein disulfide isomerase that limits UPR signaling, resulted in increased XBP1 splicing during UPR abatement, accompanied with increased sensitivity to ER stress[39]. While in agreement with the literature, it is currently unclear why, in *isw1Δ* cells, deficient UPR abatement leads to incomplete stress mitigation and decreased cell viability upon ER stress[3,4,13], although UPR induction appears indistinguishable from WT cells. An appealing explanation is that sustained UPR activation overloads the translocation machinery[4,40], leading to mislocalization of ER resident proteins, some of which having crucial functions in ER homeostasis. This is supported by our Pdi1 glycosylation time-courses. Upon Tm wash, inhibition of protein glycosylation is released and coincides with the appearance of five discrete bands above deglycosylated Pdi1. These likely correspond to the mono, di, tri, tetra and penta glycosylated forms of the protein that is translocated post-translationally[41] and would receive the N-glycan on each of its consensus Asn residue, as it emerges into the ER lumen. The delayed Pdi1 glycosylation observed in *isw1Δ* cells would thus be symptomatic of a saturation of the translocation apparatus that can virtually concern a wide-range of ER-destined proteins, ultimately leading to increased sensitivity to ER stress. Alternatively, this could result from the activation of cell death pathway(s) triggered upon persistent Ire1/Hac1 signaling. This would be reminiscent of mammals where the UPR initially engages adaptive outputs to lessen the load of unfolded proteins, but which, under chronic or unresolved ER stress causes cells commitment to apoptosis[2].

ISWI factors were previously implicated in the regulation of stress responses. Yeast Isw1, in combination with other chromatin remodelers, has been associated with the regulation of expression of heat stress genes[42,43]. Similarly, homologs of Isw1 were reported to contribute to stress-induced memory of heat shock genes in plants[44] or to control the expression of cytosolic chaperones in response to mitochondrial stress in *C. elegans*[45]. In all these circumstances however, the influence of ISWI on the stress responses are mediated by a transcriptional effect whereas the Isw1-mediated UPR regulation that we report relies on its RNA binding activity.

Strikingly, the subcellular localization of *HAC1* mRNA proves to be crucial for accurate UPR signaling. Seminal work from Peter Walter's laboratory demonstrated that a 3′BE contained in *HAC1* 3′UTR is essential to its targeting to Ire1 foci[31]. Our data unveil that the 3′UTR carries additional localization signals: the IBM1 motif, whose interaction with Isw1 allows fine-tuning the rate of *HAC1* transcript export to the cytoplasm. Given the previously reported low basal activity of Ire1[46,47], this nuclear export control appears essential to prevent excessive or untimely *HAC1* mRNA splicing, as observed in *isw1Δ* cells in the absence of stress (Fig. 2b, c and ref. 46. The ability of over-expressed Isw1 to shape the UPR, and especially to rescue the phenotypes of *ire1D828A* cells, shores up the critical function of the Isw1-mediated *HAC1* mRNA retention in accurate UPR attenuation and is lending support to the prominent role of mRNA nuclear retention on the regulation of gene expression. With respect to the UPR, this activity of Isw1 is remarkable for it highlights a previously unrecognized level of regulation of the signaling abatement, that emanates from the nucleus and acts synergistically with the previously described Ire1 deactivation-mediated pathways of UPR attenuation. Although poly(A) RNA nuclear retention was acknowledged more than 40 years ago[48], its potential significance for gene expression has long been ignored. It drew growing attention over the last decade and was shown to serve various functions[49,50], such as quality control, rapid adaptation to stress, regulation of protein level, or more recently to provide a general mean to buffer "noisy" gene expression resulting from bursts in gene transcription[51,52]. Because *HAC1* mRNA is exported from the nucleus in WT conditions, its interaction with Isw1 is expected to be weak or transient and outcompeted by the binding of nuclear export factors, therefore only "putting a break" on the kinetics of its nuclear export. While we observed an increase in the Isw1-*HAC1* mRNA interaction upon ER stress due to *ISW1* UPR induction, it is unknown whether the affinity of Isw1 for RNA is regulated upon specific conditions.

Given the prominence of the intensity and duration of the UPR for cell fate decisions under ER stress, ISWI/SMARCA5, whose RNA binding activity is conserved in *Drosophila*[53] and mammalian cells[54,55], appears

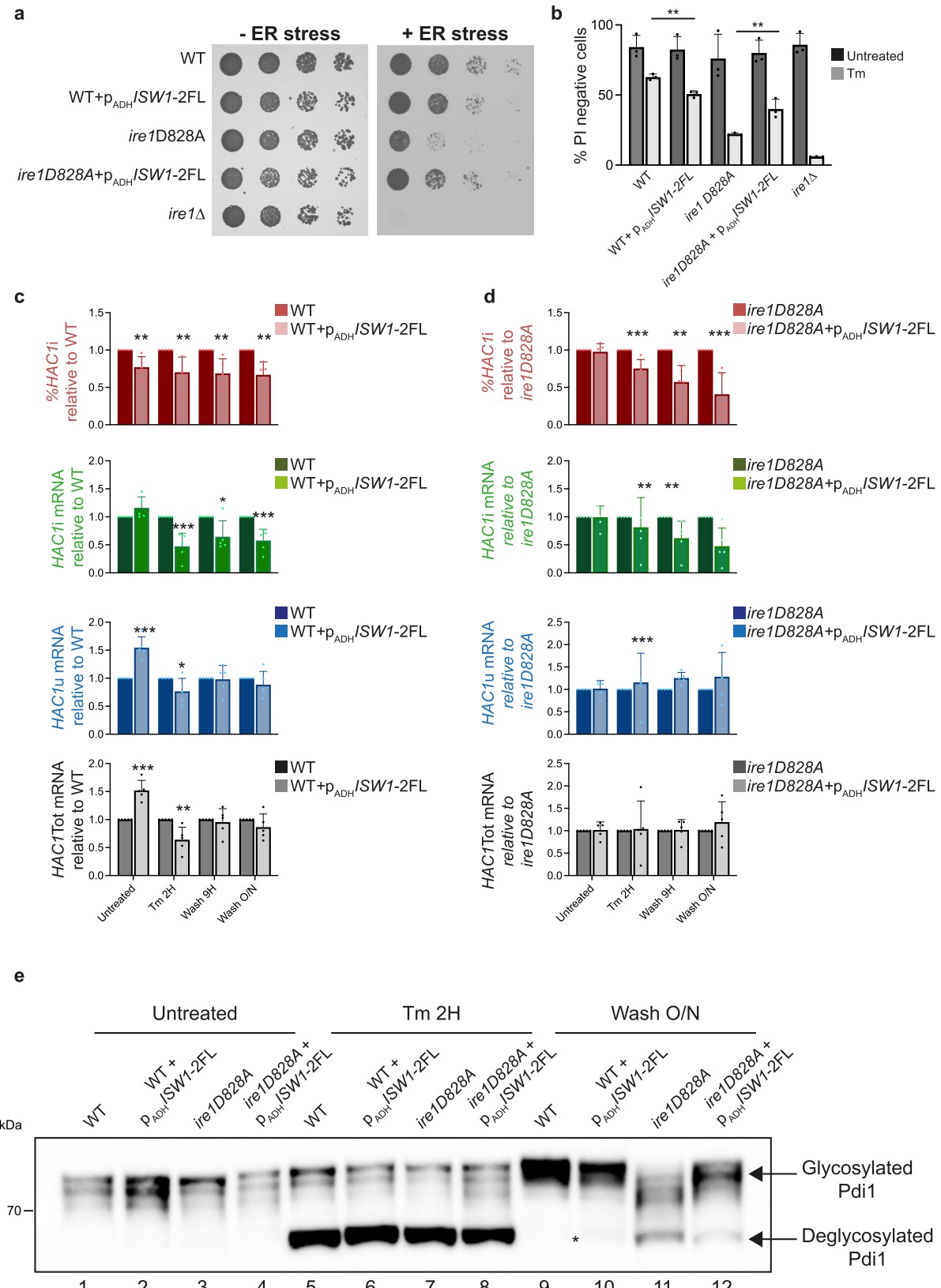

as a promising target for therapeutic intervention against the growing number of human diseases associated with ER stress and inappropriate UPR regulation[56,57]. Interestingly, GWA studies associated a SNP yielding a truncated mutant of human SMARCA5 to Amyotrophic lateral sclerosis[58], a disease with multiple connections to ER stress and RNA metabolism[59,60].

## Methods

### Yeast strains

All experiments were performed using *Saccharomyces cerevisiae* strains of the W303 background. Strains were grown at 30 °C according to classical procedures in standard yeast extract peptone dextrose (YPD) or dropout base supplemented with the appropriate CSM-amino

**Fig. 6 | Isw1 overexpression negatively regulates the UPR. a** Isw1 overexpression sensitizes WT cells to Tm but partially rescues the growth of *ire1D828A* cells on Tm. Fivefold serial dilutions of the indicated strains grown for 3 days at 30 °C on selective media containing (+ER stress) or not (−ER stress) 0,4 µg/mL Tm. **b** Viability of the indicated strains was analyzed as described in Fig. 3b. $n = 3$ independent experiments, mean ± sd. Unpaired one-tailed *t*-tests (*p* values under Tm conditions: WT + $p_{ADH}$*ISW1*−2FL relative to WT:1.28E−03, *ire1D828A* + $p_{ADH}$*ISW1*−2FL relative to *ire1D828A:* 5.67E−03). **c, d** Isw1 overexpression impairs *HAC1* mRNA splicing. qRT-PCR analysis of the expression levels of *HAC1*i, *HAC1*u, and *HAC1*Tot relative to WT in the indicated strains. $n = 5$ independent experiments, mean ± s.d. Unpaired one-tailed *t*-tests (*p* values %*HAC*i for WT + $p_{ADH}$*ISW1*−2FL relative to WT: 3.23E−03, 5.78E−03, 3.62E−03 and 1.34E−03 for Untreated, Tm 2H, Wash 3H and Wash O/N conditions respectively). *p* values %*HAC*i for *ire1D828A* + $p_{ADH}$*ISW1*−2FL + $p_{ADH}$*ISW1*−2FL relative to *ire1D828A*: 3.25E−0, 8.76E−04, 1.27E−03 and 8.75E−04 for Untreated, Tm 2H, Wash 3H and Wash O/N conditions respectively). *p* values of *HAC1*i, *HAC1*u and *HAC1*Tot are shown in the source data file. Unnormalized values shown in Supplementary Fig. 7b. **c, e** Isw1 overexpression improves Pdi1 re-glycosylation in *ire1D828A* cells during Tm UPR time course. Total protein extracts from WT and *ire1D828A* cells overexpressing or not Isw1 and collected during Tm UPR time courses were analyzed by western blot with anti-Pdi1 antibodies. *indicates the persistence of deglycosylated Pdi1. Three independent experiments were performed with similar results.

acid (MP Biomedicals, LCC) media. To induce ER stress, yeast cells were treated with 2 mM of dithiothreitol (DTT, VWR) or 1 µg/ml of tunicamycin (Tm, Sigma). Gene deletions and tagging were created by one-step PCR mediated strategy, as described in ref. [61]. The *HAC1*- 3′ UTR *ACT1* chimeric strain was built by insertion of a KanMx cassette 283 bp downstream of *ACT1*, followed by PCR amplification of a 3′UTR *ACT1*-KanMx product flanked by 50 bp sequences surrounding *HAC1* 3′ UTR, that was transformed into the WT strain for replacement of the endogenous *HAC1* 3′UTR. IBM1 (positions 39−135 of *HAC1* 3′UTR) and IBM2 (300−428) deletions were generated by mutagenesis (Q5® Site directed mutagenesis, NEB) in pRS316-*HAC1* and introduced genomically by PCR with a KanMx cassette. All mutations were verified by sequencing. Yeast cells used in this study are described in Supplementary Table 1.

## Plasmids

pRS416$_{IOC4}$-*ISW1-2FL* plasmid was built by replacing the endogenous *ISW1* promoter in plasmid pRS416-*ISW1-2FL* by the *IOC4* promoter (603 bp region upstream of ATG) amplified from WT W303 strain, via homologous recombination. The plasmids were verified by sequencing.

Plasmids used in this study are described in Supplementary Table 2.

## Plate-based growth assays

Cells inoculated in the appropriate medium and grown overnight at 30 °C were diluted to OD$_{600}$ = 0.4 grown for 2 h and re-diluted to OD$_{600}$ = 0.8 in water. Fivefold serial dilutions were plated with a Replica plater. Plates were imaged after 3 days of growth at 30 °C. When indicated, tunicamycin was added to the growth medium at concentrations ranging from 0.2 to 0.8 µg/mL. Control plates contained 4 µL of DMSO (drug carrier). Each presented growth assay is a representative example of at least three biological replicates.

## Viability assay

Cells were suspended in PBS and stained with 5 µg/mL propidium iodide (1 mg/mL stock solution, dissolved in water) for 20 min in the dark at room temperature. PI fluorescence was examined from at least 400 cells in three biological replicates under a fluorescence microscope and red fluorescent cells (dead) were counted.

## Chromatin immunoprecipitation

Cells were cross-linked in the presence of 1% formaldehyde for 10 min at 25 °C. After quenching with 100 mM glycine and washing with Tris-buffered saline, cell pellets were resuspended in 1 ml lysis buffer (50 mM Hepes pH 7.5, 140 mM NaCl, 1 mM EDTA, 1% triton X-100, 0.1% deoxycholate, 1X protease inhibitors cocktail, complete EDTA-free, Roche) and lysis was achieved in the presence of acid washed glass beads (Sigma) using the MagNAlyser (Roche). Cell lysates were collected and sonicated using a Bioruptor (Diagenode) to shear DNA for three runs of 10 min. Solubilized chromatin was retrieved through a 5 min-centrifugation at 2500 × *g*. Anti-RNA

Polymerase II antibodies (OptimAb™ RNA Polymerase II 8WG16, Eurogentec) were added and immunoprecipitation was performed by overnight rotation at 4 °C. Protein-G sepharose beads (GE Healthcare) were added to the samples for two additional hours. Washes were as follow: twice with lysis buffer, twice with lysis buffer containing 360 mM NaCl; twice with 10 mM Tris pH 8, 250 mM LiCl, 0.5% Nonidet-P40, 0.5% deoxycholate, 1 mM EDTA and once with 10 mM Tris−HCl pH 8, 1 mM EDTA. Elution was performed through a 20-min incubation at 65 °C in the presence of 50 mM Tris pH 8, 10 mM EDTA, 1% SDS. The eluate was deproteinized with proteinase K (Sigma, 0.2 mg/mL) and uncrosslinked for 30 min at 65 °C. Immunoprecipitated DNA was purified with the Qiaquick PCR purification kit (Qiagen) and quantified by real-time PCR with a Quant-Studio5™ (Applied Biosystems™) according to the manufacturer's instructions.

## qRT-PCR analysis and RNA immunoprecipitation

For qRT-PCR analysis, RNA was isolated from 10 OD of exponentially growing cells washed in MilliQ water and snap frozen. RNA was extracted with the Nucleospin RNA II Kit (Macherey-Nagel), and reverse transcribed using the SuperScript II reverse transcriptase (Invitrogen). cDNA was quantified by qPCR (Master Mix PCR Power SYBR™ Green ThermoFisher; QuantStudio5™; Applied Biosystems™). Analysis of *HAC1* mRNA splicing was performed as follows. Primers qAB 88/qAB91, qAB135/91 and qAB124/125 were designed to specifically amplify *HAC1*u, *HAC1*i and *HAC1*Tot respectively, with qAB88 spanning the exon1-intron junction and qAB135 spanning the exon-exon junction of the transcript. Specificity of primers qAB88 and qAB135 was validated by qRT-PCR analysis performed on a mix of two plasmids bearing the unspliced and spliced form of *HAC1* with the following ratios: u/s = 1/0-0.75/0.25-0.5/0.5- 0.25/0.75-0/1. The percentage of *HAC1* mRNA splicing was assessed using the LEM-PCR method[62]. For the three amplicons (*HAC1*u, i, Tot), standard curves were generated from the same cDNA derived from the WT Tm 2H samples. The relative abundance of *HAC1*u, *HAC1*i and *HAC1*Tot were used to derive two linear equations:

$$\text{Untreated}: x.HAC1u_{\text{Untreated}} = y.HAC1i_{\text{Untreated}} + HAC1\text{Tot}_{\text{Untreated}} \quad (1)$$

$$\text{Tm2H}: x.HAC1u_{\text{Tm2H}} = y.HAC1i_{\text{Tm2H}} + HAC1\text{Tot}_{\text{Tm2H}} \quad (2)$$

The *x* and *y* values determined by solving these equations were substituted into the equation for each sample and allowed to determine the percentage of spliced *HAC1* during UPR time-courses.

RNA immunoprecipitations were performed from 2 g of frozen cell grindates prepared as previously described[26]. Immunoprecipitated RNA was isolated from proteins by treating samples with 40 µg Proteinase K (Roche) and 0.1% SDS for 30 min at 30 °C. RNA was extracted with the Nucleospin RNA II Kit (Macherey-Nagel), and reverse transcribed using the SuperScript II reverse transcriptase (Invitrogen).

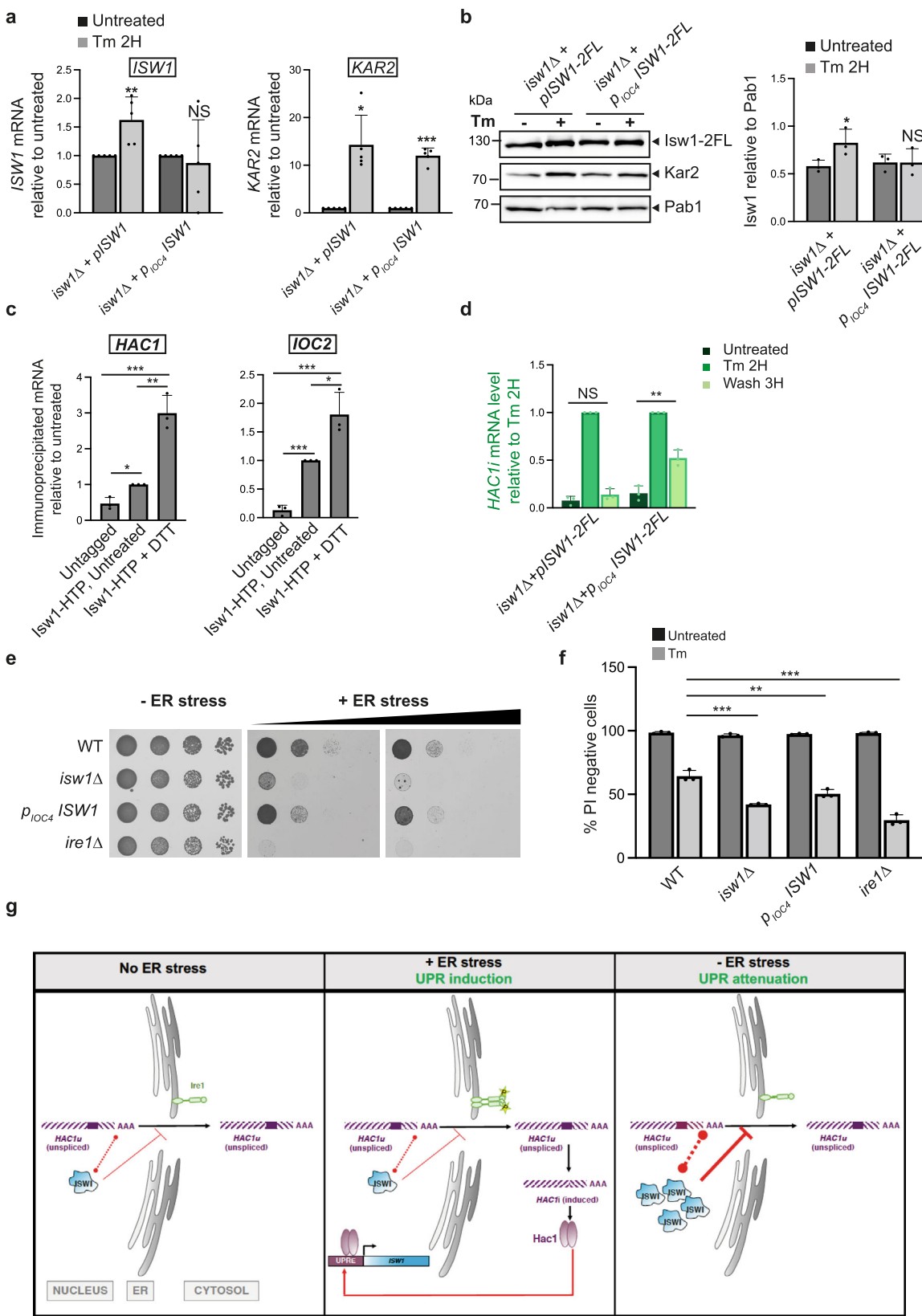

cDNA was quantified by qPCR as previously described. Primers used in this study are described in Supplementary Table 3.

## Deep sequencing

mRNA sequencing libraries were prepared from polyA+ RNAs purified from 1µg of total RNA using NEBNext® UltraTM RNA Library Prep Kit for Illumina® (NEB, USA) following the manufacturer's recommendations. A 150 bp read sequencing was Illumina sequenced with multiplexing. A mean of 17.7 ± 2.4 million passing Illumina quality filter reads was obtained for each of the samples. Library preparation and sequencing with Illumina technology were performed by Novogene (Novogene Co., Ltd).

**Fig. 7 | UPR-mediated *ISW1* induction allows a negative feedback control of the pathway. a** Swapping of *ISW1* endogenous promoter prevents its ER stress induction. Tm-induced *ISW1* expression is abolished in cells expressing $p_{IOC4}ISW1$ while *KAR2* induction is unaffected, as evaluated by qRT-PCR analysis. $n = 3$ independent experiments, mean ± s.d. Unpaired one-tailed *t*-tests (*p* values relative to Untreated for *ISW1* target: 4.22E−03 and 3.83E−01 for *isw1Δ*+p*ISW1*−2FL and *isw1Δ*+$p_{IOC4}ISW1$−2FL respectively. *p* values relative to Untreated for *KAR2* target: 6.94E−04 and 1.70E−07 for *isw1Δ*+p*ISW1*−2FL and *isw1Δ*+$p_{IOC4}ISW1$−2FL respectively). **b** Swapping of *ISW1* endogenous promoter prevents the induction of Isw1 by Tm. Total protein extracts from *isw1Δ* cells expressing *pISW1-2FL* or *isw1Δ*+$p_{IOC4}ISW1$-2FL grown in the presence (+) or absence (−) of 1 µg/mL Tm for 2 H were analyzed by western blot with anti-FLAG, anti-Pab1 (loading) and anti-Kar2 (UPR induction) antibodies and the level of Isw1 relative to Pab1 was quantified. n = 3 independent experiments, mean ± s.d. Unpaired one-tailed *t*-tests (*p* values Tm 2 H relative to Untreated: 2.69E−02 and 4.96E−01 for *isw1Δ*+*pISW1-2FL* and *isw1Δ*+$p_{IOC4}ISW1$-2FL cells respectively). **c** The interaction between Isw1 and its mRNA targets is fostered by ER stress. RIP assays in Isw1-HTP cells treated or not with 2 mM DTT for 1 H. Ratio of co-immunoprecipitated RNA to total RNA relative to untreated quantified by qRT-PCR. *n* = 3 independent experiments, mean ± s.d. Unpaired one-tailed *t*-tests (*p* values relative to Untreated for *HAC1* target: 2.77E−03 and 1.12E−03 for Untagged and Isw1-HTP + DTT respectively. (*p* values relative to Untreated for *IOC2* target: 3.21E−05 and 1.18E−02 Untagged and Isw1-HTP + DTT respectively. *p*

values relative to Untagged for *HAC1* target: 2.77E−03 and 5.65E−04 for Isw1-HTP, Untreated and Isw1-HTP + DTT respectively). *p* values relative to Untagged for *IOC2* target: 3.21412E−05 and 9.68E−04 for Isw1-HTP, Untreated and Isw1-HTP + DTT respectively). **d** Cells expressing non-UPR-inducible *ISW1* ($p_{IOC4}ISW1$) are defective for UPR termination. *HAC1i* normalized to *SCR1* mRNA expression during Tm UPR time courses, relative to Tm 2 H, as evaluated by qRT-PCR in *isw1Δ* cells expressing *pISW1* or *isw1Δ*+$p_{IOC4}ISW1$. $n = 3$ independent experiments, mean ± s.d. Unpaired one-tailed *t*-tests (*p* values Wash 3 H relative to Untreated: 1.10E−01 and 2.56E−03 for *isw1Δ*+*pISW1-2FL* and *isw1Δ*+$p_{IOC4}ISW1$-2FL cells respectively. Unnormalized values shown in Supplementary Data Fig. 8a. **e** Cells deficient for UPR-mediated *ISW1* induction are more sensitive to Tm than WT. Fivefold serial dilutions of the indicated strains grown in the presence (+ER stress) or absence (−ER stress) of 0.4 µg/mL Tm. **f** Viability of the indicated strains was analyzed as described in Fig. 3b. $n = 3$ independent experiments, mean ± s.d. Unpaired one-tailed *t*-tests (*p* values relative to WT under Untreated conditions: 4.46E−01, 5.25E−02 and 3.31E−01 for *isw1Δ*, $p_{IOC4}$ *ISW1* and *ire1Δ* cells respectively. *p* values relative to WT under Tm conditions: 5.19E−04, 6.18E−03 and 3.11E−04 for *isw1Δ*, $p_{IOC4}$ *ISW1* and *ire1Δ* cells respectively). **g** Model: Cross-talk between Isw1-mediated nuclear mRNA quality control and the Unfolded protein response. The nuclear export of *HAC1* mRNA is restricted by direct binding of the chromatin remodeler Isw1 to its 3'UTR. This nuclear retention activity of Isw1 is reinforced by its UPR induction and appears critical for accurate UPR termination and cell survival upon ER stress.

## CRAC assay

CRAC has been performed according to ref. [63] with the modifications described in ref. [64]. Briefly, 2 L of yeast cells expressing Isw1-HTP tag or Rpb-HTP tag were grown at 30 °C to $OD_{600} = 0.6$ in CSM-Trp medium. Cells were UV crosslinked using a W5 UV crosslinking unit (UVO3 Ltd) for 50 s, harvested by centrifugation, washed in cold PBS and resuspended in TN150 buffer (50 mM Tris pH 7.8, 150 mM NaCl, 0.1% NP-40 and 5 mM β-mercaptoethanol, 2.4 mL/g of cells) supplemented with protease inhibitors (Complete, EDTA-free Protease Inhibitor Cocktail). The suspension was flash frozen in droplets and cells were mechanically broken with a Mixer Mill MM 400. Extracts were treated for one hour at 25 °C with DNase I (165 U/g of cells) to solubilize chromatin and clarified by centrifugation. Complexes were purified by a two-step procedure, the second one under denaturing conditions. Both purification steps were performed under high stringency (1 M NaCl). The protein fractionation step was performed with a Gel Elution Liquid Fraction Entrapment Electrophoresis (GelFree) system (Expedeon). Isw1-containing fractions were treated with 100 mg of proteinase K. RNAs were purified and reverse transcribed with Superscript IV (Invitrogen). The concentration of cDNAs in the reaction was estimated by quantitative PCR and accordingly PCR amplified. After treatment with 200 U/ml of Exonuclease I (NEB), DNA was purified using NucleoSpin Gel and PCR Clean-up (Macherey-Nagel) and sequenced using Illumina technology.

## Next-generation sequence analysis

For RNA-Seq analysis, raw data were processed through in-house scripts. Paired-end clean reads were mapped to the reference genome using HISAT2 software. Reads were aligned to the *S. cerevisiae* reference genome (SaCer3). HTSeq software was used to analyze the gene expression levels in this experiment, using the union mode. Differential expression analysis between two conditions (three biological replicates per condition) was performed using DESeq2, R package. The resulting *P* values were adjusted using the Benjamini and Hochberg's approach for controlling the false discovery rate (FDR). Genes with an adjusted *P* value < 0.05 found by DESeq2 were assigned as differentially expressed.

CRAC datasets were analyzed as described[28]. The pyCRAC script pyFastqDuplicateRemover was used to collapse PCR duplicates using a 6 nucleotides random tag included in the 3′ adaptor. The resulting sequences were reverse complemented with Fastx reverse complement (part of the fastx toolkit, http://hannonlab.cshl.edu/fastx_toolkit/) and

mapped to the R64 genome[65] with bowtie2 (−N 1)[66]. In order to identify with higher precision the binding position a further analysis was performed to extract deletions and mismatches inside the mapped reads and collect their positions in a coverage file. This step was performed with the peakCcall repository[64].

## Protein extracts and immunoblotting

Cells grown in the appropriate medium to $OD_{600}$ 0.4 were treated with 1 µg/ml of Tm for 2 h when indicated. 20% trichloroacetic acid (TCA, Sigma) was added to the culture. 10 OD of cells were collected for each point, pelleted and washed in 20% TCA at 4 °C. Pellets were snap frozen in liquid nitrogen and further processed by cells resuspension in 200 µl of TCA 20% and disruption by vortexing for 10 mn at 4 °C in the presence of 100 µl of acid washed glass beads (Sigma). After beads removal, samples were centrifuged at $11,000 \times g$ for 10 min and pellets were resuspended in Laemmli buffer, heated for 5 min at 95 °C. Protein were separated by SDS-PAGE and transferred onto a nitrocellulose membrane that was probed with the relevant antibody. Protein were detected by chemiluminescence (SuperSignal™ West Pico/Femto Chemiluminescent Substrate, ThermoFisher) and images were captured with the FUSION FX imaging system (Vilber). Antibodies used in this study are described in Supplementary Table 4.

## Fluorescence in situ hybridization

FISH was performed as previously described[26,67] with minor modifications. Cells grown in the appropriate media were fixed by adding parafolmaldehyde to the media at the final concentration of 4%. Fixation was performed for 45 min at room temperature (RT) on a rotating wheel. After 2 washes in wash buffer (100 mM KHPO4, pH 7.5), cells were pelleted and resuspended in spheroplast buffer (1.2 M sorbitol, 100 mM $KHPO_4$, pH 7.5) supplemented with 0,5 mg/mL 100 T zymolyase and 2 mM vanadyl-ribonucleoside complex. Digestion was performed for approximately 15 min at 30 °C (until cell wall digestion of >80% of the cells). Spheroplasts were carefully washed twice in cold spheroplast buffer, resuspended in 70% ethanol and stored for at least 12 h at 4 °C. After cell rehydratation by resuspension in spheroplast buffer for 15 min at RT, spheroplasts were hybridized in hybridization buffer (Formamide 50%, Dextrane sulfate 10%, 4X SSC, 1X Denhardts, 125 mg/ml E. coli tRNA, 500 mg/ml salmon sperm DNA) supplemented with 2 ng of *HAC1*-Quasar 670 probes and 2 mM vanadyl-ribonucleoside for 12 h in the dark at 30 °C. The next day, spheroplasts were washed twice in 2XSSC at 30 °C for one hour and 1XSSC

for 30 min at room. DNA was stained with Hoechst 33342 (Sigma) and spheroplasts mounted onto mounting media. All coverslips were finally washed in 1XPBS plus Hoechst and mounted onto Vectashield© antifade mounting media.

The *HAC1* FISH probe was purchased from Biosearch technologies and contains a blend of 27 oligos spanning *HAC1* exons labeled with Quasar®670. Probe specificity was verified against a *hac1Δ* strain. Quantifications were performed on maximum intensitiy z projections. For quantification of nuclear signal intensities, nuclei were segmented from the Hoechst images, the overlapping *HAC1* Quasar®670 signals were determined using the "image calculator" function of the ImageJ software and measured with the "Analyze particles" function. Cytoplasmic signal intensities were measured in the same cells from random cytoplasmic circular ROI (1 μm radius).

## Fluorescence microscopy

Three-dimensional stacks with a 0.2-μm step were acquired by 3D deconvolution microscopy using a Axiovert 200 M microscope (Carl Zeiss MicroImaging, Inc.) with a 63X oil immersion objective (NA = 1.4). Images were captured using a monochrome digital camera (Axiocam MRm; Carl Zeiss MicroImaging, Inc). Maximum intensity projections were performed using ImageJ software.

## Statistics and reproducibility

Every depicted blot or growth assay is representative of at least three independent and reproducible experiments.

Plots and statistical analysis were performed using Excel (Microsoft) for qRT-PCR analysis or Prism (Graphpad) for RNAseq analysis and RNA FISH quantification. Statistic tests and *p* values are described in the figure legends. Significance of the observed differences (*$p$ value 0.01–0.05; **$p$ value 0.001–0.01; ***$p$ value <0.001). For each experiment, the value of every data point as well as $p$ values are provided in the source data file.

## Reporting summary

Further information on research design is available in the Nature Research Reporting Summary linked to this article.

## Data availability

Unique biological material generated in this study is available from the corresponding author upon reasonable request. Source data for each figure are provided with this paper as a source data file. Gene expression data have been deposited in ArrayExpress under accession code E-MTAB-10511. Rpb1 CRAC has been deposited to GEO under the accession code GSE207652. Source data are provided with this paper.

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

## Acknowledgements

We wish to thank Madhusudan Dey, Maho Niwa, Michel Tolédano and Peter Walter for generous reagents sharing. We are grateful to Benoît Palancade, Vincent Géli, Michel Tolédano, for critical reading of the manuscript as well as stimulating discussions. This work was supported by the Agence Nationale pour la Recherche (ANR-JC, grant 17-CE12-0023-01), the Fondation ARC pour la recherche sur le cancer (PJA 20181207794 to AB), the Ligue Contre le Cancer (RS19/75-45), the LabEx "Who Am I?" (#ANR-11-LABX-0071) and the Université de Paris (IdEx #ANR-18-IDEX-0001) funded by the French Government through its "Investments for the Future" program. The initial CRAC experiments were supported by the LabEx "Who am I?" as a "collaborative project grant" between DL and AB (then part of C. Dargemont's team). This work has benefited from the facilities and expertise of the high throughput

sequencing core facility of I2BC (Centre de Recherche de Gif - http://www.i2bc-saclay.fr/).

## Author contributions

Conceptualization, A.B.; Methodology, A.B., D.C., M.B., and L.M.; Investigation, A.B., D.C., L.M., and M.B.; Formal analysis A.B. and L.M.; Writing, A.B.; Visualization, A.B. and L.M.; Funding acquisition, A.B. and D.L.; Supervision, A.B. and D.L.

## Competing interests

The authors declare no competing interests.

## Additional information

**Correspondence and requests** for materials should be addressed to Anna Babour.

