## [Peer Review File · Nature Communications]

Termination of the unfolded protein response is guided by ER stress-induced HAC1 mRNA nuclear retentionREVIEWER COMMENTS

Reviewer #1 (Remarks to the Author):

Key results

The unfolded protein response (UPR) contains multiple layers of regulation to enable cells to both respond to and recover from stress. This manuscript demonstrates the HAC1 mRNA, which encodes a central transcription factor which activates the UPR, is bound by the yeast chromatin remodeler Isw1, itself a transcriptional target of HAC1, through a moonlighting function for Isw1 in mRNP quality control. The authors show that *isw1Δ* yeast are sensitive to ER stress, and that this sensitivity is due to failure to accurately attenuate the UPR, without affecting HAC1 mRNA stability or the activity of the nuclease that splices this mRNA (*Ire1*) to activate the UPR. The authors characterize reduced levels of HAC1 mRNA in the nucleus of *isw1Δ* cells as well as reduced glycosylation of an ER resident chaperone protein. They define the precise sites of Isw1 binding within the HAC1 mRNA's 3'-UTR via CRAC experiments and demonstrate that removal of an Isw1 binding motif phenocopies the *isw1Δ* mutant. Of note, overexpression of Isw1 provided a partial rescue of yeast mutant with a molecularly distinct downstream defect in UPR attenuation due to inability to inactivate *Ire1* splicing of HAC1. Together, these results describe another layer of regulation in the UPR pathway, and highlight how transient changes in mRNA retention can be used to modulate gene expression and fine-tune cellular homeostasis.

Comments

While the authors have shown that the UTR element is responsible for the effect, I'm left wondering what the role of the 3'-BE previously described by Aragon et al. is in this process. The new elements appear to flank the BE; what is the significance of the positioning in the UTR? Additional experiments focused on the BE are essential to rule in or out what role the BE plays in this new model. Minimally, a construct containing IBM elements but lacking the BE should be examined for nuclear retention / UPR attenuation; previously the Δ BE was shown to be tunicamycin sensitive, so such an analysis could help delineate roles of Isw1 in UPR activation versus attenuation.

The authors have convincingly demonstrated that these results cannot be explained by previously characterized mechanisms for activating or attenuating the UPR. However, by only focusing on the interaction between Isw1 and the HAC1 mRNA, I'm left wondering if perhaps some of these results could be explained by Isw1 binding other transcriptional targets of HAC1, especially given the result in 7c that ER stress increases the amount of multiple mRNAs that co-IPs with Isw1. A more careful presentation of the specificity of Isw1 binding is important more confidence that the mechanism is due to direct Isw1-Hac1 binding and not additional targets of Isw1.

The heavy reliance on qRT-PCR makes it challenging to understand the magnitude of change in HAC1 splicing / localization affected by *Isw1* disruption. Particularly in the context of the rescue experiment in Figure 6b-c (and associated extended data 7b-c), the relative changes in spliced HAC1 that enable this partial rescue appear to be quite subtle, but it's unclear whether this is reflective of actual mRNA levels.

Discussion of PDI glycosylation results could be more clear. The anti-PDI westerns can only comment on the overall glycosylation status of the PDI chaperone protein, not whether it has been de- or re-glycosylated. Is there a deglycosylase present *in vivo* to act on this protein, or just in the *in vitro* experiment? If not, then this could be more accurately described as glycosylated and not-yet-glycosylated protein. The text should be expanded to explain the underlying biology in more detail and the wording could be improved to glycosylated and non-glycosylated PDI, dropping mention of re-glycosylation.

Reviewer #2 (Remarks to the Author):

In this manuscript Matabishi-Bibi et al. show intriguing data on how chromatin remodeling factor *Isw1* regulates the termination of ER stress response in *Saccharomyces cerevisiae* through nuclear retention of HAC1 mRNA. This work is a nice follow-up to the report describing how *Isw1* functions as a quality control factor of nuclear mRNP biogenesis (PMID: 27863241) and provides previously unknown mechanism on how chromatin remodeling factor regulates stress response. Although experiments are well performed, my concern is that can such conclusions be made from the data. I have listed below my comments that I think need to be considered in order to publish the manuscript.

Major:

1. Fig. 2b-2c. HAC1i expression is increased upon loss of *isw1* also in non-stressed cells. Moreover, the expression of HAC1i transcriptional targets is increased in *isw1* mutant cells in non-stressed conditions (Fig. 2e, extended Figs. 2a and 2g), thus indicating that *isw1* mutants have activated UPR already in basal conditions. Therefore, authors should show the analysis of RNA-seq for UPR targets in basal conditions. Could delayed UPR termination after tunicamycin treatment in *isw1* mutant cells be due to the basal ER stress, and therefore, return to homeostasis after additive stress (*Tm* treatment) requires extended stress response?

2. Fig. 2c. Authors show % HAC1i mRNA relative to wild type. Similar graph should be shown from HAC1u and HAC1Tot. This will show whether *isw1* mutation affects HAC1 expression.

3. Fig. 4., extended Fig. 5. Although *isw1* mutation reduces the nuclear export of HAC1 mRNA, it is still abundant in the cytoplasm of *isw1* mutant cells. I left wondering whether such a small difference can explain the delay in UPR termination? This will be difficult to address experimentally. However, it would be interesting to see the ratio between nuclear and cytoplasmic HAC1 mRNA in both WT and *isw1* mutants. This would improve the reader's understanding about the magnitude of *Isw1*-mediated retention of nuclear HAC1 mRNA.

4. Does *Isw1* overexpression further increase the level of nuclear HAC1 mRNA in WT cells? This should be checked with HAC1 FISH.

5. In the Discussion authors agree that it is unclear why *isw1* mutant cells are more sensitive to ER stress although the induction of ER stress response is similar to WT cells. Authors should show whether general health of *isw1* mutant cells is affected (e.g. lifespan and sensitivity to oxidative and mitochondrial stress), which may also cause increased susceptibility to external stress (Tm in this case).

6. Fig. 6. Like *isw1* mutation, also *Isw1* overexpression increases susceptibility to ER stress in WT cells. If this phenotype is due to the inhibited HAC1 splicing, and since authors state that *Isw1* and *Ire1* regulate UPR through two independent pathways, why *Isw1* overexpression does not further decrease the survival of *ire1D828A* mutants on Tm?

7. Fig. 7a-b. Why *isw1* expression upon Tm treatment is measured from *isw1*-overexpressing cells and not from WT cells? Authors should also show the expression of endogenous *isw1* upon ER stress.

Minor:

8. Although similar observation has been done earlier, it is perplexing to reader how impaired termination of ER stress response reduces cell survival upon ER stress. This could be discussed more in the paper.

9. There are many reports on how ISW factors regulate different stress responses in various organisms. Authors could discuss this, which would also support the proposed role of *Isw1* in ER stress.

Reviewer #3 (Remarks to the Author):

Report on manuscript "Termination of the unfolded protein response is guided from the nucleus by ER stress-induced HAC1 mRNA nuclear retention", by L. Matabishi-Bibi et al.

This manuscript reports a role for the ISW1 chromatin remodeling complex in the UPR response by retaining unspliced HAC1 mRNAs in the nucleus. Indeed, HAC1 mRNAs encode a key transcriptional regulator of the UPR response and undergo an unusual cytoplasmic splicing catalyzed by the IRE1 endonuclease and the tRNA ligase Tlg1. In this work the authors use a large array of experiments to show that ISW1 directly binds HAC1 mRNAs and increase its nuclear retention, and that deletion of ISW1 or its binding site on HAC1 mRNAs slow down abatement of the UPR response after ER stress removal. The interest of the paper lies in the demonstration of the binding of the chromatin regulator ISW1 to HAC1 mRNA and the functional consequences of this binding. Indeed, interactions between RNA and chromatin remodelers have been reported in various model systems but we still have a poor understanding of the function of these interactions. The present paper makes thus an important advance.

Overall, the study is interesting and well done with carefully crafted experiments. I have two general criticisms and several more precise points.

Major points:

1-The effects reported are generally small and this dampens my enthusiasm for the paper as it may reduce its significance. For instance, swapping the 3' UTR of HAC1 mRNA (where ISW1 binds) with that of ACT1 reduces ISW1 binding by only two-fold (fig 1b); the increase in spliced HAC1 mRNA and HAC1 protein in ISW1^Δ strain during recovery is also 2 fold (fig 2c,d); the effect on HAC1 mRNA localization in the nucleus and the cytoplasm by smFISH are less than two-fold in the nucleus and very small in the cytoplasm (fig 4bc, extended data figure 5); the effect of the HAC1^ΔIBM1 mRNA mutants (which binds less ISW1) in cell viability during ER stress seems small (fig 5b). While the role of ISW1 on resistance to ER stress is clear (fig 3a), the proposed nuclear retention mechanism may contribute only partially to this phenotype.

2-The role of ISW1 is not entirely clear to me. Indeed, it seems to me that most, if not all experiments could be interpreted in two different manners. First, ISW1 may dampen the UPR response during stress recovery. Second, ISW1 may promote UPR response during stress. For instance, in fig 2b where HAC1 mRNA is measured, the mRNA may be more abundant at later time point after Tm removal in ^ΔISW1 cells because the response could be less efficient in absence of ISW1 and may thus require longer to recover. Even if HAC1i mRNA levels reach a similar level, other UPR genes may be affected by removing

ISW1. The effect of ISW1 deletion of Pdi glycosylation would in fact go into this direction: tunicamycin inhibit glycosylation and after its removal it takes longer to fully reglycosylate Pdi in absence of ISW1, suggesting that the stress lasts longer without ISW1. In the FISH experiment, the nuclear HAC1 transcription dot may be less intense without ISW1 because there is less transcription, not because the mRNA is released. ISW1 indeed has been involved in maintaining proper chromatin during transcription (PMID: 22922743 and others), so a positive transcriptional effect is quite possible.

I think that the authors should perform a genome wide transcription assay (such as PolII ChIP) +/- ISW1 +/- ER stress, to see how this chromatin remodeler affect transcription of HAC1 and the UPR genes.

Minor comments.

1-Introduction: it would be better to add few lines to precise the function of ISW1 on chromatin and transcription.

2-The authors identified HAC1 mRNA as a direct ISW1 RNA target but the CRAC experiment on which this is based is not shown. An extract of the data is shown in extended fig 1, but in absence of the full dataset the meaning of this figure is unclear as one cannot judge how this binding compares to other RNA or other proteins. There is in addition no unit for the y axis. A full (but brief) report of the CRAC data should be shown.

3- Fig2, why does the spliced HAC1 mRNA increases in ISW1 mutants during recovery ? if more RNA are released while stress is gone, shouldn't this be the unspliced ?

4-For the RNA seq experiment (fig 2f), it would be useful to also plot WT vs ISW1^Δ at different time points (untreated, stress, recovery), to see if some mRNAs are particularly affected by ISW1 deletion.

5-For the FISH experiments, I could not figure out what the authors are measuring in the cytoplasm. The simplest and more meaningful metric would be to simply plot the number of cytoplasmic mRNAs per cell. In the nucleus, the values could be expressed in number of RNA molecules rather than arbitrary units. The authors must calculate the fraction of nuclear RNA (# nuclear RNA vs total RNA), this is very important as it seems that the amount of retained RNA is small.

6-In fig 5b, 6a, 7e, the authors should provide doubling time for the indicated strains as this is a more quantitative assay and the differences in the plate growth assay appear small.

We would like to thank the reviewers for underlining the originality of our work, which reveals a previously unappreciated mechanism of UPR regulation and highlights a physiological role for the direct interaction between the chromatin remodeler Isw1 and one of its mRNA target, *HAC1*. We appreciate their very constructive and helpful comments as well as the overall positive perception of the manuscript. We have now addressed these comments and hope that this significantly improved version of the manuscript now meets the editorial criteria required for publication.

Reviewer 1 (Remarks to the Author):

Key results

The unfolded protein response (UPR) contains multiple layers of regulation to enable cells to both respond to and recover from stress. This manuscript demonstrates the *HAC1* mRNA, which encodes a central transcription factor which activates the UPR, is bound by the yeast chromatin remodeler Isw1, itself a transcriptional target of *HAC1*, through a moonlighting function for Isw1 in mRNP quality control. The authors show that *isw1* Δ yeast are sensitive to ER stress, and that this sensitivity is due to failure to accurately attenuate the UPR, without affecting *HAC1* mRNA stability or the activity of the nuclease that splices this mRNA (*Ire1*) to activate the UPR. The authors characterize reduced levels of *HAC1* mRNA in the nucleus of *isw1* Δ cells as well as reduced glycosylation of an ER resident chaperone protein. They define the precise sites of Isw1 binding within the *HAC1* mRNA's 3'-UTR via CRAC experiments and demonstrate that removal of an Isw1 binding motif phenocopies the *isw1* Δ mutant. Of note, overexpression of Isw1 provided a partial rescue of yeast mutant with a molecularly distinct downstream defect in UPR attenuation due to inability to inactivate *Ire1* splicing of *HAC1*. Together, these results describe another layer of regulation in the UPR pathway, and highlight how transient changes in mRNA retention can be used to modulate gene expression and fine-tune cellular homeostasis.

Comments

1. While the authors have shown that the UTR element is responsible for the effect, I'm left wondering what the role of the 3'-BE previously described by Aragon et al. is in this process. The new elements appear to flank the BE; what is the significance of the positioning in the UTR? Additional experiments focused on the BE are essential to rule in or out what role the BE plays in this new model. Minimally, a construct containing IBM elements but lacking the BE should be examined for nuclear retention / UPR attenuation; previously the Δ BE was shown to be tunicamycin sensitive, so such an analysis could help delineate roles of Isw1 in UPR activation versus attenuation.

We would like to thank reviewer 1 for this very relevant comment. The 3'BE region, responsible for targeting the *HAC1* mRNA to *Ire1* foci for optimal splicing, is indeed flanked by the two IBM motifs that we identified. While the deletion of IBM1 -contrary to that of IBM2- does not perturb UPR activation, we have not examined whether deletion of IBM1/IBM2 and the 3'BE had overlapping effects on *HAC1* mRNA nuclear export and targeting to *Ire1*.

We have now analyzed and compared the UPR transcriptional output, the Tm sensitivity, and the subcellular localization of the *HAC1* transcript in cells inactivated for the 3'BE or co-inactivated for the 3'BE and for *ISW1*, as well as in WT and *isw1* Δ

cells. Altogether, these results indicate that although IBM1 and the 3'BE are located close to one another in *HAC1* 3'UTR, they carry distinct functions in UPR regulation.

These new results are now presented in Fig. S6 d, e, f.

Text has been modified as follow:

line 234

Finally, comparison of the UPR transcriptional output, the Tm sensitivity, and the subcellular localization of the *HAC1* transcript in WT, *isw1Δ*, *hac1Δ3'BE* ($\Delta 3'BE$) and *isw1Δ*, *hac1Δ3'BE* cells reinforced the distinction between the 3'BE and the IBM regions. As previously reported, $\Delta 3'BE$ cells were defective for *HAC1* mRNA splicing and sensitive to ER stress²⁹ (Supplementary Fig. 6d, e). They showed delayed *HAC1*i production -presumably resulting from a compensatory increased expression of total *HAC1* (Supplementary Fig. 6d)- and unperturbed *HAC1* mRNA subcellular localization (Supplementary Fig. 6f). Co-inactivation of *ISW1* and the 3'BE resulted in increased *HAC1*Tot and prolonged *HAC1*i expression, reduced *HAC1* mRNA nuclear localization compared to WT and increased Tm sensitivity of the double mutant compared to each single mutant (Supplementary Fig. 6 d, e, f), implying that both mutations act in different processes. Taken together, these results indicate that the phenotype of *isw1Δ* cells is recapitulated by deletion of the *HAC1* 3'UTR-located IBM1 sequence, that is distinct from the 3'BE, and that mediates the interaction of the *HAC1* transcript with *Isw1*.

2. The authors have convincingly demonstrated that these results cannot be explained by previously characterized mechanisms for activating or attenuating the UPR. However, by only focusing on the interaction between *Isw1* and the *HAC1* mRNA, I'm left wondering if perhaps some of these results could be explained by *Isw1* binding other transcriptional targets of *HAC1*, especially given the result in 7c that ER stress increases the amount of multiple mRNAs that co-IPs with *Isw1*. A more careful presentation of the specificity of *Isw1* binding is important more confidence that the mechanism is due to direct *Isw1*-*Hac1* binding and not additional targets of *Isw1*.

Isw1 binds indeed many RNAs, some of which code for UPR targets. It is therefore possible that some of these *Isw1* targets, in addition to *HAC1*, might contribute to the UPR phenotype of *isw1Δ* cells. However, the CRAC output by itself would not be sufficiently reliable for identifying strong *Isw1* binders as potential UPR actors without additional experiments. This is because the signal depends on the levels of transcription (if *Isw1* binding is co-transcriptional) or RNA steady-state (if *Isw1* binding is post-transcriptional) and we cannot reliably estimate the co-transcriptional *versus* post-transcriptional share of *Isw1* binding. Therefore, we opted for using CRAC results as a starting point that identified *HAC1* as a putative *Isw1* target and designed further experiments to validate this hypothesis. Because preventing the binding of *Isw1* to *HAC1* mRNA phenocopies to a large extent the deletion of *ISW1*, we consider that *HAC1* is the main effector of *Isw1* in UPR. We cannot exclude a role of other UPR factors that could also be retained by *Isw1*, but their validation, even having established they are bound by *Isw1*, would require the generation of analogous mutants that would be likely associated with more subtle phenotypes. We consider that these experiments would not be compatible with a reasonable revision time frame. However, for complying with the referee's request, we show a qualitative analysis of *Isw1* binding to all genes, with the UPR genes highlighted in Figure S1b. In this stringent analysis we

have retained only the Isw1 peaks that are not found in the no-crosslinked control, regardless of their intensity (i.e we have also excluded the peaks that are quantitatively different in the two sets).

These new analysis are shown in Supplementary Fig1.b

The text has been modified as follow:

line 76:

While the full genome wide analysis of these experiments is beyond the scope of this report, we have identified that Isw1 binds the *HAC1* mRNA (Supplementary Fig.1a) , among others (Supplementary Fig.1b), and analyzed the corresponding reads.

line 299:

Our CRAC experiment demonstrate, as previously reported²⁶, the binding of Isw1 to multiple mRNA species whose nuclear export are expected to be limited by Isw1. While we cannot exclude that these putative Isw1 targets might also impact UPR signaling, we assume that Isw1 binding to *HAC1* mRNA has stronger physiological consequences because it not only regulates the level of the transcript in the cytoplasm but also affects its splicing and thereby the synthesis of the short-lived Hac1 protein.

3. The heavy reliance on qRT-PCR makes it challenging to understand the magnitude of change in *HAC1* splicing / localization affected by Isw1 disruption. Particularly in the context of the rescue experiment in Figure 6b-c (and associated extended data 7b-c), the relative changes in spliced *HAC1* that enable this partial rescue appear to be quite subtle, but it's unclear whether this is reflective of actual mRNA levels.

In Fig. 2b-2c, RT-qPCR was used to reveal sustained *HAC1* splicing in *isw1Δ* cells. This, was accompanied with sustained Hac1 protein expression, as evaluated by Western blot (Fig 2d). Similarly, we now show that the overexpression of Isw1 that leads to reduced *HAC1* splicing (Fig. 6 c) is accompanied with reduced production of the Hac1 protein. These new results are shown in Supplementary Fig. 7d.

Text has been modified as follows:

line 262

Monitoring the transcriptional UPR output in these cells during Tm-induced UPR time-courses revealed that Isw1 overexpression correlated with a significant reduction of *HAC1* splicing in WT - in line with a significant reduction of Hac1 expression (Supplementary Fig. S7e) - and *ire1D828A* cells (Fig. 6c, d).

In addition, we now show, as advised by reviewer 2 (see point 4), that Isw1 overexpression significantly increases the steady-state nuclear localization of the *HAC1* transcript. This is now presented in Supplementary Fig 7d.

Text has been modified as follows:

line 253

We therefore overexpressed Isw1 in WT or *ire1D828A* cells (Supplementary Fig. 7a), which led to an increased nuclear localization of the *HAC1* transcript (Supplementary Fig. 7b).

4. Discussion of PDI glycosylation results could be more clear. The anti-PDI westerns can only comment on the overall glycosylation status of the PDI chaperone protein, not whether it has been de- or re- glycosylated. Is there a deglycosylase present *in vivo* to act on this protein, or just in the *in vitro* experiment? If not, then this could be more accurately described as glycosylated and not-yet-glycosylated protein. The text should be expanded to explain the underlying biology in more detail and the wording could be improved to to glycosylated and non-glycosylated PDI, dropping mention of re-glycosylation.

We apologize for the poor phrasing and lack of clarity and thank the reviewer for pointing it out. The N-glycosidase, PNGase, was used here only *in vitro* to demonstrate that the slow-migrating Pdi1 band appearing upon cell's treatment with tunicamycin corresponded to the non-glycosylated form of Pdi1. There is, to our knowledge, no endogenous glycosidase. N-linked protein glycosylation, in the ER, involves the assembly of an oligosaccharide on a lipid carrier and the transfer of the oligosaccharide to selected asparagine residues of polypeptides that have entered the lumen of the ER. As Tm is washed out, N-glycosylation is re initiated. During the wash period, in addition to the de-glycosylated form of the protein, we observe 5 discrete bands, which may represent the mono, di, tri, tetra and penta-glycosylated forms of Pdi1 (each N-glycan adding approximately 2,5 kDa to the protein). This would indeed be consistent with the fact that Pdi1 is translocated in the ER post-translationally in an exclusively SRP-independent manner (PMID: 8707814). As such, newly synthesized full length Pdi1 is progressively modified on its consensus Asn residues as it is translocated into the ER lumen. In *isw1Δ* cells, complete Pdi1 glycosylation is delayed compared to WT, likely reflecting a perturbed translocation into the ER. This is in line with previous studies (PMID: 21444684; PMID: 30865586) reporting that prolonged ER stress leads to defective ER protein translocation resulting from overloading of the translocation machinery.

The text has been modified as follow:

line 144

Pdi1 features N-glycosylation at five sites²⁷, which can be abolished upon deglycosylation with PNGase, resulting in increased mobility of the protein by SDS PAGE (Extended data Fig. 3b). When glycosylation was inhibited for two hours with tunicamycin, Pdi1 was detected as two bands, corresponding to the glycosylated and unglycosylated forms of the protein. Tunicamycin induced partial Pdi1 deglycosylation in both WT and *ire1Δ* cells. In WT cells, upon removal of the drug and during the abatement phase of the UPR, Pdi1 progressively recovered its initial glycosylation status. In *ire1Δ* cells, Pdi1 remained deglycosylated, reflecting the inability of cells that are unable to initiate the UPR to alleviate the stress (Extended data Fig. 3c). Strikingly, upon identical experimental settings, the kinetics of Pdi1 re-glycosylation...

line 308:

While in agreement with the literature^{3,4,13}, it is currently unclear why, in *isw1Δ* cells, deficient UPR abatement leads to incomplete stress mitigation and decreased cell viability upon ER stress although UPR induction appears indistinguishable from WT cells. ~~It may be that the resulting sustained production of ER molecular chaperones sequesters client proteins with critical functions for ER homeostasis.~~ An appealing explanation is that sustained UPR activation overloads the translocation machinery^{4,37}, leading to mislocalization of ER resident proteins, some of which having crucial functions in ER homeostasis. This is supported by our Pdi1 glycosylation time-courses. Upon Tm wash, inhibition of protein glycosylation is released and coincides with the appearance of five discrete bands above deglycosylated Pdi1. These likely correspond to the mono, di, tri, tetra and penta glycosylated forms of the protein that is translocated post-translationally³⁸ and would receive the N-glycan on each of its consensus Asn residue, as it emerges into the ER lumen. The delayed Pdi1 glycosylation observed in *isw1Δ* cells would thus be symptomatic of a saturation of the translocation apparatus that can virtually concern a wide-range of ER-destined proteins, ultimately leading to increased sensitivity to ER stress. Alternatively,...

Reviewer #2 (Remarks to the Author):

In this manuscript Matabishi-Bibi et al. show intriguing data on how chromatin remodeling factor *Isw1* regulates the termination of ER stress response in *Saccharomyces cerevisiae* through nuclear retention of *HAC1* mRNA. This work is a nice follow-up to the report describing how *Isw1* functions as a quality control factor of nuclear mRNP biogenesis (PMID: 27863241) and provides previously unknown mechanism on how chromatin remodeling factor regulates stress response. Although experiments are well performed, my concern is that can such conclusions be made from the data. I have listed below my comments that I think need to be considered in order to publish the manuscript.

Major

1. Fig. 2b-2c. *HAC1i* expression is increased upon loss of *isw1* also in non-stressed cells. Moreover, the expression of *HAC1i* transcriptional targets is increased in *isw1* mutant cells in non-stressed conditions (Fig. 2e, extended Figs. 2a and 2g), thus indicating that *isw1* mutants have activated UPR already in basal conditions. Therefore, authors should show the analysis of RNA-seq for UPR targets in basal conditions. Could delayed UPR termination after tunicamycin treatment in *isw1* mutant cells be due to the basal ER stress, and therefore, return to homeostasis after additive stress (Tm treatment) requires extended stress response?

Inactivation of *ISW1* indeed results in a slight increase in the basal level of *HAC1* mRNA splicing in the absence of ER stress. This is presumably caused by the increased nuclear export of the transcript which becomes cytoplasmically spliced by the basal activity of *Ire1* (see also answer to Reviewer 3 minor 3), leading to UPR activation without proteostasis defect, as indicated by similar Pdi1 glycosylation profiles observed in WT and *isw1Δ* cells in the absence of Tm. Hence, increased cytoplasmic localization of *HAC1* mRNA resulting from *ISW1* inactivation allows its splicing and results in basal low level of UPR activation in the absence of exogenous ER stress and persistent *HAC1* splicing upon treatment with ER stress-inducing agents. Although our RT-qPCR analysis reveals a modest but significant increase in

HAC1 splicing in agreement with a previous report (PMID: 19325107), genome-wide DE-seq analysis of UPR genes expression revealed no significant difference between WT and *isw1Δ* cells under basal conditions, in agreement with the literature (PMID: 12482963, PMID: 21596317). The analysis of RNA-seq comparing WT to *isw1Δ* cells for the expression of UPR target genes is now shown for each point of the time-course (Untreated, Tm 2H, Wash 3H) in Supplementary Fig S2d. Significant differences between both strains are only observed during the recovery phase of the UPR (Wash 3H), arguing against a delayed UPR activation upon *ISW1* inactivation.

The manuscript has been modified as follow :

line 115

As such, differential gene expression between WT and *isw1Δ* cells was only observed 3 hours after Tm removal (Supplementary Fig 2d).

2. Fig. 2c. Authors show % HAC1i mRNA relative to wild type. Similar graph should be shown from HAC1u and HAC1Tot. This will show whether *isw1* mutation affects HAC1 expression.

We have now added to Figure 2b, graphs showing the relative expression of each form of the *HAC1* transcript.

3. Fig. 4., extended Fig. 5. Although *isw1* mutation reduces the nuclear export of HAC1 mRNA, it is still abundant in the cytoplasm of *isw1* mutant cells. I left wondering whether such a small difference can explain the delay in UPR termination? This will be difficult to address experimentally. However, it would be interesting to see the ratio between nuclear and cytoplasmic HAC1 mRNA in both WT and *isw1* mutants. This would improve the reader's understanding about the magnitude of *Isw1*-mediated retention of nuclear HAC1 mRNA.

We previously discovered the RNA-binding activity of *Isw1* in the context of mRNA export mutants in which *ISW1* inactivation was able to release nuclear retained transcripts to the cytoplasm. The effects we are looking at here are indeed more subtle: *HAC1* is abundant in the cytoplasm in a WT context and *ISW1* inactivation increases this cytoplasmic localization. We would like to thank the reviewer for suggesting to show ratios between nuclear and cytoplasmic *HAC1* mRNA, which indeed clarifies our message.

We have now represented the quantification of *HAC1* mRNA FISH experiments as a ratio of nuclear to cytoplasmic *HAC1* mRNA signal in figures 4c, S5 b, c, 5f and S7b. The text was modified accordingly:

line 197

This analysis uncovered a significant decrease in *HAC1* nuclear signal intensity, mirrored by an increase in the cytoplasmic *HAC1* mRNA signal , recapitulated by a significant decrease in the ratio of nuclear to cytoplasmic *HAC1* mRNA signal intensity, in *isw1Δ* compared to WT cells (Fig. 4c), indicating that the absence of *Isw1* correlates with a reduction of the steady-state nuclear localization of *HAC1* transcript.

4. Does *Isw1* overexpression further increase the level of nuclear *HAC1* mRNA in WT cells? This should be checked with *HAC1* FISH.

We have now analyzed *HAC1* mRNA localization by RNA FISH in WT cells (*isw1Δ* + *pISW1*) and cells overexpressing *ISW1* (*isw1Δ* + *p_{ADH} ISW1*). We indeed observe an increase in the nuclear to cytoplasmic *HAC1* signal ratio upon *ISW1* overexpression. This is in line with our previously published result, showing that *ISW1* overexpression aggravates the phenotype of a *np13-1* mutant -whose growth is defective due to compromised mRNA nuclear export- as a result of an increased *Isw1*-mediated nuclear mRNA retention (PMID: 27863241). This new result is presented in Supplementary Fig 7b.

Text has been modified as follows:

line 252

We therefore overexpressed *Isw1* in WT or *ire1D828A* cells (Supplementary Fig. 7a) which led to an increased nuclear localization of the *HAC1* transcript (Supplementary Fig. 7b).

5. In the Discussion authors agree that it is unclear why *isw1* mutant cells are more sensitive to ER stress although the induction of ER stress response is similar to WT cells. Authors should show whether general health of *isw1* mutant cells is affected (e.g. lifespan and sensitivity to oxidative and mitochondrial stress), which may also cause increased susceptibility to external stress (Tm in this case).

The drastic consequences of ER stress on the viability of *isw1Δ* cells - that are nonetheless able to mount a UPR similar to that of WT cells - is indeed intriguing. We now discuss what we think is the most likely explanation for why *isw1Δ* cells are more sensitive to ER stress than WT (See answer to reviewer 1 point 4 and reviewer 2 point 8). Interestingly, this sensitivity to ER stress appears to be quite specific as we did not observe any increased sensitivity of *isw1Δ* cells compared to WT to oxidative, osmotic or mitochondrial stress. These new results are presented in Figure 3d and S3b. The text has been modified as follows:

line 139

This stress sensitivity phenotype was not general, as no difference in sensitivity to oxidative, osmotic or mitochondrial stress was observed between WT and *isw1Δ* cells (Fig. 3d, Supplementary Fig.3b).

A number of large-scale screen for yeast genes whose deletion affects chronological lifespan (CLS) have been previously published. In particular, *ISW1* inactivation was reported to lead to a weak (approximately 3 days) increase in CLS (PMID: 15722108) or to have no influence on CLS (PMID: 20657825), implying that *Isw1* is not a key factor controlling yeast lifespan.

Altogether, these data do not support the idea of a general weakness of *isw1Δ* compared to WT cells, but rather a specific role for *Isw1* in the regulation of the cell's response to ER stress.

6. Fig. 6. Like *isw1* mutation, also *Isw1* overexpression increases susceptibility to ER stress in WT cells. If this phenotype is due to the inhibited *HAC1* splicing, and since authors state that *Isw1* and *Ire1* regulate UPR through two independent pathways, why *Isw1* overexpression does not further decrease the survival of *ire1D828A* mutants on Tm?

ISW1 overexpression limits *HAC1* mRNA splicing in both WT and *ire1D828A* cells. In WT cells, this has a (slight) negative effect on cell viability in the presence of ER stress. In *ire1D828A* cells, whose hypersensitivity to ER stress results from defective UPR attenuation, *HAC1* splicing is modulated in opposite directions by the *ire1D828A* mutation (increases splicing duration) and the overexpression of *ISW1* (limits splicing). The increase in viability observed upon *ISW1* overexpression in this context suggests that *ISW1* overexpression is sufficient to partially overcome the detrimental effects of *ire1D828A* mutation.

7. Fig. 7a-b. Why *isw1* expression upon Tm treatment is measured from *isw1*-overexpressing cells and not from WT cells? Authors should also show the expression of endogenous *isw1* upon ER stress.

We apologize for this misunderstanding. In fact, in Fig 7a-b, cells do not overexpress *ISW1*. The expression of *Isw1* is measured from *isw1Δ* cells transformed with either *pISW1-2FL* which expresses *ISW1* under control of its own promoter (800 pb) at its endogenous level or with *p_{IOC4} ISW1-2FL* which results from the swapping of *ISW1* endogenous promoter for that of *IOC4* in *pISW1-2FL* plasmid. We can also detect the ER stress induction of *Isw1* from cells expressing a genomically tagged version of the protein as shown below.

ER stress induction of endogenously tagged *Isw1*

a. WT cells expressing an endogenously FLAG tagged version of *ISW1* were treated (+) or not (-) with 1ug/mL tunicamycin for 2H. Total protein extracts were analyzed by Western blot with anti-FLAG, anti-Kar2 and anti-Pab1 antibodies.

b. Same as a., full membrane. * is a non-specific band detected by the anti-FLAG antibody.

The legend in figure 7 has been modified as follows for clarification:

a Swapping of *ISW1* endogenous promoter prevents its ER stress induction

Minor

8. Although similar observation has been done earlier, it is perplexing to reader how impaired termination of ER stress response reduces cell survival upon ER stress. This could be discussed more in the paper.

We entirely agree that the consequences of persistent UPR signaling on cell viability are puzzling. We think it may result from defective protein translocation into ER as previously proposed (PMID: 21444684; PMID: 30865586), which is supported by our experiments analyzing Pdi1 glycosylation. See answer to reviewer 1 point 4 for a more detailed explanation as well as text modifications in the discussion section.

9. There are many reports on how ISW factors regulate different stress responses in various organisms. Authors could discuss this, which would also support the proposed role of Isw1 in ER stress.

We have now expanded the discussion with different illustrations of the role of ISWI factors in stress responses. The text has been modified as follow:

line 327:

ISWI factors were previously implicated in the regulation of stress responses. Yeast Isw1, in combination with other chromatin remodelers, has been associated with the regulation of expression of heat stress genes^{39,40}. Similarly, homologs of Isw1 were reported to contribute to stress-induced memory of heat shock genes in plants⁴¹ or to control the expression of cytosolic chaperone in response to mitochondrial stress in *C. elegans*⁴². In all these circumstances however, the influence of ISWI on the stress responses are mediated by a transcriptional effect whereas the Isw1-mediated UPR regulation that we report relies on its RNA binding activity.

Reviewer #3 (Remarks to the Author):

Report on manuscript "Termination of the unfolded protein response is guided from the nucleus by ER stress-induced HAC1 mRNA nuclear retention", by L. Matabishi-Bibi et al.

This manuscript reports a role for the ISW1 chromatin remodeling complex in the UPR response by retaining unspliced HAC1 mRNAs in the nucleus. Indeed, HAC1 mRNAs encode a key transcriptional regulator of the UPR response and undergo an unusual cytoplasmic splicing catalyzed by the IRE1 endonuclease and the tRNA ligase Tlg1. In this work the authors use a large array of experiments to show that ISW1 directly binds HAC1 mRNAs and increase its nuclear retention, and that deletion of ISW1 or its binding site on HAC1 mRNAs slow down abatement of the UPR response after ER stress removal. The interest of the paper lies in the demonstration of the binding of the chromatin regulator ISW1 to HAC1 mRNA and the functional consequences of this binding. Indeed, interactions between RNA and chromatin remodelers have been reported in various model systems but we still have a poor understanding of the function of these interactions. The present paper makes thus an important advance.

Overall, the study is interesting and well done with carefully crafted experiments. I have two general criticisms and several more precise points.

Major points:

1-The effects reported are generally small and this dampens my enthusiasm for the paper as it may reduce its significance. For instance, swapping the 3' UTR of HAC1 mRNA (where ISW1 binds) with that of ACT1 reduces ISW1 binding by only two-fold (fig 1b); the increase in spliced HAC1 mRNA and HAC1 protein in ISW1 Δ strain during recovery is also 2 fold (fig 2c,d); the effect on HAC1 mRNA localization in the nucleus and the cytoplasm by smFISH are less than two-fold in the nucleus and very small in the cytoplasm (fig 4bc, extended data figure 5); the effect of the HAC1 Δ IBM1 mRNA mutants (which binds less ISW1) in cell viability during ER stress seems small (fig 5b). While the role of ISW1 on resistance to ER stress is clear (fig 3a), the proposed nuclear retention mechanism may contribute only partially to this phenotype.

While the effects of inactivation of *ISW1* on the measured phenotypes are relatively small, they stand nonetheless in a range comparable to what has been previously reported for mutants affecting UPR attenuation. As an example, inactivation in human cells of PDIA6, a protein disulfide isomerase that limits UPR signaling (PMID: 24508390), was shown to prolong XBP1 splicing during UPR abatement (Fig 3A) -with an increase in XBP1 splicing similar to what is observed for *HAC1* upon *ISW1* inactivation- and to increased cell sensitivity to ER stress by (Fig 2C).

In addition, as also mentioned by reviewer 1 (point 2), it remains possible that *lsw1*-mediated *HAC1* mRNA nuclear retention is only one component of the effect of *lsw1* on UPR attenuation. *lsw1* binding to other ER stress-induced transcripts could also participate in the observed effects. See answer to reviewer 1 point 2.

The text has been modified as follows

line 305

Although the effect of *ISW1* inactivation on UPR signaling may seem modest, it stands in a similar range to that of previous reports. As such, inactivation of PDIA6, a protein disulfide isomerase that limits UPR signaling, results in increased XBP1 splicing during UPR abatement, accompanied with increased sensitivity to ER stress³⁹

2-The role of ISW1 is not entirely clear to me. Indeed, it seems to me that most, if not all experiments could be interpreted in two different manners. First, ISW1 may dampen the UPR response during stress recovery. Second, ISW1 may promote UPR response during stress. For instance, in fig 2b where HAC1 mRNA is measured, the mRNA may be more abundant at later time point after Tm removal in Δ ISW1 cells because the response could be less efficient in absence of ISW1 and may thus require longer to recover. Even if HAC1 mRNA levels reach a similar level, other UPR genes may be affected by removing ISW1. The effect of ISW1 deletion of Pdi glycosylation would in fact go into this direction: tunicamycin inhibit glycosylation and after its removal it takes longer to fully reglycosylate Pdi in absence of ISW1, suggesting that the stress lasts longer without ISW1. In the FISH experiment, the nuclear HAC1 transcription dot may be less intense without ISW1 because there is less transcription, not because the

mRNA is released. ISW1 indeed has been involved in maintaining proper chromatin during transcription (PMID: 22922743 and others), so a positive transcriptional effect is quite possible. I think that the authors should perform a genome wide transcription assay (such as PolII ChIP) +/- ISW1 +/- ER stress, to see how this chromatin remodeler affect transcription of HAC1 and the UPR genes.

The phenotype of *isw1Δ* cells could result from an impaired -and as a result delayed and prolonged- UPR activation, similarly to what we observe for the *hac1Δ3BE* mutant (now shown in Supplementary Fig. S6) that is defective for *HAC1* mRNA splicing. However, data from the literature, reporting, under basal growth conditions, very similar expression profiles (PMID: 21596317, PMID: 12482963, PMID: 21940898) or RNA Pol II ChIP profiles (PMID: 26861626, Spearman's rank correlation coefficient 0,961) between WT and *isw1Δ* cells do not favor this hypothesis. In addition, our transcriptomic analysis of WT and *isw1Δ* cells during Tm induced UPR time-courses does not show any significant differential gene expression neither under untreated conditions nor upon 2H Tm induction but only during the abatement phase of the stress response (Supplementary Fig S2d), supporting a role for Isw1 in the termination phase of the UPR rather than in the transcriptional induction of the pathway.

In order to definitely clarify this aspect, we compared RNA Pol II occupancy by ChIP qRT-PCR on the 25 most robustly expressed UPR genes between WT and *isw1Δ* cells, under untreated growth conditions or after a 2H Tm treatment. No significant difference in RNA Pol II recruitment to UPR genes was detected between WT and *isw1Δ* cells, neither under basal conditions nor upon stress induction. These novel data are presented in Supplementary Fig. S2e.

We also, as suggested by the referee, performed a genome-wide transcriptional assay in WT and *isw1Δ* cells, under normal growth conditions or after a 2H Tm treatment, using Rpb1 CRAC. Using this sensitive technique, we found a minor effect of *ISW1* deletion on *HAC1* expression as well as a minor genome-wide effect when the median Rpb1 signal is plotted on all genes or on UPR target genes. The results of this novel experiment are now presented in Supplementary Fig. S2f,g.

The text has been modified as follows

line 116:

In addition, to exclude the possibility that the sustained UPR activation observed in *isw1Δ* results from an impaired and therefore delayed UPR activation, we conducted a careful comparison of UPR induction after a 2 hours Tm treatment in WT and *isw1Δ* cells. The analysis of RNA Polymerase II (RNAPII) recruitment on 25 of the most robustly induced UPR target genes by ChIP-qPCR (Supplementary Fig 2e), as well as the genome-wide profiling of UPR transcription using the CRAC technique to detect the position of RNAPII on UPR targets, revealed that WT and *isw1Δ* cells display comparable UPR induction profiles (Supplementary Fig 2f-g).

Minor comments:

1. Introduction: it would be better to add few lines to precise the function of ISW1 on chromatin and transcription.

The text has been modified as follows:

line 54

... which we recently reported to be initiated at the chromatin level by the nucleosome spacing-enzyme Isw1. This ATP-dependent chromatin remodeler contributes to the regular spacing and phasing of the nucleosomes over coding regions¹⁷. Although inactivation of *ISW1* leads to global perturbation of chromatin organization -associated with increased intragenic cryptic transcription¹⁸, and characterized by reduced nucleosome spacing¹⁹⁻²¹ - it has little influence on cell growth and gene expression, typified by a modest derepression of very few genes^{22,23,25,26}. Strikingly however, it promotes the release from chromatin and consecutive export of nuclear-retained mRNPs²⁷.

2-The authors identified *HAC1* mRNA as a direct ISW1 RNA target but the CRAC experiment on which this is based is not shown. An extract of the data is shown in extended fig 1, but in absence of the full dataset the meaning of this figure is unclear as one cannot judge how this binding compares to other RNA or other proteins. There is in addition no unit for the y axis. A full (but brief) report of the CRAC data should be shown.

As detailed in the answer to reviewer 1 point 2, we now present a whisker plot showing the overall CRAC signal (log2) contained in the specific peaks of Isw1 binding to its RNA targets in Supplementary Fig S1.b . Supplementary Fig S1.a now depicts the distribution of the sites of crosslink of Isw1 to the *HAC1* mRNA.

In addition, the binding of *HAC1* mRNA to Isw1 appears to be in the same range than its binding to the Cap Binding Protein subunit Cpb20. Indeed, Fig.4d reports on the binding of *HAC1* to Cpb20 as evaluated by RIP and shows a fold enrichment (over untagged) of approximately x7 while the fold enrichment of *HAC1* binding to Isw1 is approximately x4 (Fig. 5d).

3. Fig 2, why does the spliced *HAC1* mRNA increases in ISW1 mutants during recovery ? if more RNA are released while stress is gone, shouldn't this be the unspliced ?

Upon *ISW1* inactivation, more unspliced *HAC1* is indeed exported from the nucleus. However, Ire1 is known to display low basal activity upon non-ER stress conditions (PMID: 19325107), which has allowed to screen for genes whose deletion caused either up-regulation or down-regulation of the UPR, using a GFP-based UPR reporter. We interpret that the increased *HAC1* mRNA splicing observed in *isw1Δ* cells compared to WT results from nuclear release of the transcript and subsequent cytoplasmic splicing by Ire1 basal activity. This basal Ire1 activity, is believed to help supporting the maintenance of cellular homeostasis (PMID: 20513765). In particular, we previously reported in yeast cell, that basal UPR signaling facilitated cytokinesis during normal cell growth (PMID:17562790).

The following modification was made in the text:

line 339

Given the previously reported low basal activity of Ire1^{45,46}, this nuclear export control appears essential to prevent excessive or untimely *HAC1* mRNA splicing, as observed in *isw1Δ* cells in the absence of stress (Fig. 2b,c and ⁴⁵).

4. For the RNA seq experiment (fig 2f), it would be useful to also plot WT vs *ISW1Δ* at different time points (untreated, stress, recovery), to see if some mRNAs are particularly affected by *ISW1* deletion.

We now show the comparison of WT and *isw1Δ* cells at each time point of our RNA seq analysis (Untreated, Tm 2H and Wash 3H) in Fig S2d. Significant differences between both strains are only observed at Wash 3H.

The manuscript has been modified as follow :

line 115

As such, differential gene expression between WT and *isw1Δ* cells was only observed 3 hours after Tm removal (Supplementary Fig 2d).

5. For the FISH experiments, I could not figure out what the authors are measuring in the cytoplasm. The simplest and more meaningful metric would be to simply plot the number of cytoplasmic mRNAs per cell. In the nucleus, the values could be expressed in number of RNA molecules rather than arbitrary units. The authors must calculate the fraction of nuclear RNA (# nuclear RNA vs total RNA), this is very important as it seems that the amount of retained RNA is small.

We agree with the reviewer that in theory the simplest metric would be to count the number of cytoplasmic mRNAs per cell. This is indeed what we plotted in our publication reporting the function of *Isw1* in nuclear mRNA retention, using the *LYS2* transcript as a model mRNA (PMID: 27863241). However, *HAC1* mRNA is very abundant: it was ranked among the top 100 most expressed mRNA in *S. cerevisiae* and estimated at 48 copies /cell (PMID: 19040753). This prevented such an analysis as we were unable to accurately distinguish individual transcripts. Instead, as a proxy for absolute nuclear and cytoplasmic *HAC1* mRNA quantification, we quantified the intensities of the *HAC1* mRNA nuclear and cytoplasmic signals that we now represented as a nuclear to cytoplasmic ratio, as recommended by reviewer 2 (point 3).

To confirm the validity of this quantification, we have applied it to a previously published and manually quantified experiment (PMID: 27863241, Fig 5C-Fig S5B), in which the number of cytoplasmic *LYS2* transcripts was evaluated in a mutant defective for *LYS2* mRNA nuclear export (*lys2-370*) inactivated or not for *ISW1* (*lys2-370 isw1Δ*).

This new analysis is now presented in Supplementary Fig.S5b.

The text was changed as follows:

line 194:

To further strengthen this observation, we quantified, the intensities of the *HAC1* mRNA signal in the nucleus and the cytoplasm of untreated cells, an approach that we validated on a manually quantified and previously published data set²⁶ (Supplementary Fig. 5b ). This analysis uncovered a significant decrease in *HAC1* nuclear signal intensity, mirrored by an increase in the cytoplasmic *HAC1* mRNA signal, recapitulated by a significant decrease in the ratio of nuclear to cytoplasmic *HAC1* mRNA signal intensity in *isw1Δ* compared to WT cells (Fig. 4c).

6. In fig 5b, 6a, 7e, the authors should provide doubling time for the indicated strains as this is a more quantitative assay and the differences in the plate growth assay appear small.

In the growth assay presented figure 3a, in the presence of tunicamycin, *isw1Δ* cells do not present any apparent differences in the size of the colonies compared to WT cells while they show a reduction in the number of drops. This suggests that tunicamycin affects the viability of *isw1Δ* cells (number of drops) rather than their growth rate (colony size). Accordingly, as shown in the following figure, we did not detect any significant difference in the doubling time of WT and *isw1Δ* cells grown in the presence of DTT or Tm.

Doubling time WT and *isw1Δ* cells: The growth of WT and *isw1Δ* cells was monitored for 8 hours in YPD at 30°C in the presence or absence of 1mM DTT or 1 μg/mL Tm. The doubling time of the respective cultures was calculated (n=3 or 4).

In order to provide a quantitative value for the growth assays presented in figures 5b, 6a, and 7e we have now analyzed the viability (percentage of PI negative cells) of each mutant.

These new analysis are now presented Fig 5c, 6b and 7f.

REVIEWER COMMENTS

Reviewer #1 (Remarks to the Author):

The authors have adequately addressed my comments. Congratulations on a very nice study.

Reviewer #2 (Remarks to the Author):

Authors have addressed all my concerns by performing multiple new experiments and analyses as well as by editing the text. The new data supports the unexpected role of ISW1 in ER stress. I support the publication of this manuscript in Nature Communications.

Reviewer #3 (Remarks to the Author):

The authors performed a number of experiments that confirmed their initial conclusions. In particular, they add RNAPII CRAC data to show that the effect of ISW1 is post-transcriptional. I support publication of the paper and have 3 remaining comments.

1-Regarding my previous point 1, I am not totally convinced by the author's answer and I think they should be more careful in their conclusions and state clearly that other functions of ISW1 unrelated to RNA retention could contribute to stress sensitivity (on HAC1 or other genes).

2-In S2b and e, values for HAC1 should be shown.

3-I am not at all satisfied with the way the authors quantify their smFISH images. The material and methods states:

"Quantifications were performed on maximum intensity z projections. For quantification of nuclear signal intensities, nuclei were segmented from the Hoechst images, the overlapping HAC1 Quasar®670

signals were determined using the "image calculator" function of the ImageJ software and measured with the "Analyze particles" function. Cytoplasmic signal intensities were measured in the same cells from random cytoplasmic circular ROI (1 micrometer radius)."

It is NOT possible to reproduce what the authors did and also to figure out what exactly they measure.

Taking the mean value of a random cytoplasmic location is a very bad proxy to estimate a number of particles. Plus, the authors states they use the analyze particle function for the nucleus but they provide a mean intensity value in fig 4c. This is the only experiment where the authors directly measure nuclear retention, a key claim of the paper. Therefore, the authors MUST perform a proper quantification of the mRNA number in the nucleus and the cytoplasm. There are packages that do this kind of analysis in 3D and in this case 50 mRNA per cell should be perfectly tractable. Even if some particles touch and are fused during the analysis, this would still be far better than the current analysis.

We would like to thank all reviewers for the overall positive appreciation of our revised manuscript.

After discussing with all co-authors, we will be happy to provide a revised version addressing reviewer 3 comments as follows:

Reviewer #1 (Remarks to the Author):

The authors have adequately addressed my comments. Congratulations on a very nice study.

Reviewer #2 (Remarks to the Author):

Authors have addressed all my concerns by performing multiple new experiments and analyses as well as by editing the text. The new data supports the unexpected role of ISW1 in ER stress. I support the publication of this manuscript in Nature Communications.

Reviewer #3 (Remarks to the Author):

The authors performed a number of experiments that confirmed their initial conclusions. In particular, they add RNAPII CRAC data to show that the effect of ISW1 is post-transcriptional. I support publication of the paper and have 3 remaining comments.

1-Regarding my previous point 1, I am not totally convinced by the author's answer and I think they should be more careful in their conclusions and state clearly that other functions of ISW1 unrelated to RNA retention could contribute to stress sensitivity (on HAC1 or other genes).

We, accordingly, modified the text as follows (Blue = revised version 1 ; Red = revised version 2):

I 331

In all these circumstances however, the influence of ISWI on the stress responses are mediated by a transcriptional effect whereas the Isw1-mediated UPR regulation that we report mainly relies on its RNA binding activity, although we cannot exclude that other functions of ISWI could contribute to the ER stress sensitivity of *isw1*Δ cells.

2-In S2b and e, values for HAC1 should be shown.

Values for *HAC1* were added to Figures S2b and e as recommended. The source data file was accordingly modified.

3-I am not at all satisfied with the way the authors quantify their smFISH images. The material and methods states:

"Quantifications were performed on maximum intensity z projections. For quantification of nuclear signal intensities, nuclei were segmented from the Hoechst images, the overlapping *HAC1* Quasar®670 signals were determined using the "image calculator" function of the ImageJ software and measured with the "Analyze particles" function. Cytoplasmic signal intensities were measured in the same cells from random cytoplasmic circular ROI (1 micrometer radius)."

It is NOT possible to reproduce what the authors did and also to figure out what exactly they measure. Taking the mean value of a random cytoplasmic location is a very bad proxy to estimate a number of particles. Plus, the authors states they use the analyze particle function for the nucleus but they provide a mean intensity value in fig 4c. This is the only experiment where the authors directly measure nuclear retention, a key claim of the paper. Therefore, the authors MUST perform a proper quantification of the mRNA number in the nucleus and the cytoplasm. There are packages that do this kind of analysis in 3D and in this case 50 mRNA per cell should be perfectly tractable. Even if some particles touch and are fused during the analysis, this would still be far better than the current analysis.

We appreciate this remark and would like to clarify this point. As mentioned in the legend of our revised figure S5b, we quantified "the intensities of the *HAC1* mRNA nuclear and cytoplasmic signals, as a **proxy** for absolute nuclear and cytoplasmic *HAC1* mRNA quantification." For both nucleus and cytoplasm the "integrated density" of the *HAC1* mRNA signal was measured using the "analyze particles function". This is now described more precisely in the text as follows:

I 534:

For quantification of nuclear signal intensities, nuclei were segmented from the Hoechst images, the overlapping *HAC1* Quasar®670 signals were determined using the "image calculator" function of the ImageJ software and **the integrated density (product of area and mean gray value) was** measured with the "Analyze particles" function. Cytoplasmic signal intensities were **similarly** measured in the same cells from random cytoplasmic circular ROI (1 μm radius).

Because we did not have a plasma membrane staining or DIC images, it was not possible to accurately segment the cells' contour. Therefore, we quantified the signal in a random cytoplasmic circular ROI of constant size, a method that we have previously applied in PMID: 22244335, Figure 1C. In addition, we have not been able to individually quantify *HAC1* transcripts using automated packages, in line with

technical limitations of this approach, as previously described in the literature, cf. PMID: 33377036 “Still, the small size of yeast cells limit the maximum number of detected transcript to approximately 25 spots per cell, if the spots are localized in 2D and 40 spots per cell, if the spots are localized in 3D.” However, we provided in our revised version a validation of this quantification by re-quantifying a previously published and manually counted data set that reported on the effect of *ISW1* inactivation on the subcellular localization of another, less abundant, *lsw1* RNA target (Figure S5b).

Finally, this assay is not the only experiment measuring nuclear mRNA retention. We have indeed analyzed nuclear mRNPs using CBP20 RNA immunoprecipitation (Figure 4d), which revealed a significant decrease in nuclear *HAC1* mRNA upon *ISW1* inactivation, independently of RNA FISH. In addition, “transcription chase” experiments (Figure 4a-b), which examine the disappearance of *HAC1* transcription dot over-time, relies on FISH but are independent of the total signal quantification and also demonstrate that *ISW1* inactivation fosters *HAC1* mRNA nuclear export.

REVIEWERS' COMMENTS

Reviewer #3 (Remarks to the Author):

In this second round of revision, the authors have addressed two of the three remaining points. However, my previous third point, dealing with the way the smFISH images are quantified to measure cytoplasmic and nuclear RNAs, is still not addressed adequately.

The validation of their method in Figure S5b is not a good validation because it compares cytoplasmic RNA on one side and nuclear/cytoplasmic ratio on the other. It is not the same thing, and this validation does not prove that the method counts correctly nuclear RNAs. Specifically, I believe that the out-of-focus signal coming from the many cytoplasmic RNAs could confound the measurements of nuclear RNAs. Moreover, the fact that the method was used before does not make it immune to flaws and artefacts.

The author claims that traditional RNA counting, done by counting the number of spots, is not accurate enough and they use this as a justification to use a method that is then much worse. I don't see the logic here. Again, direct measurement of nuclear and cytoplasmic RNAs is an essential point of the study and the authors should therefore employ the best possible method at hand and should not compromise their efforts.

The authors already have the smFISH images and even if they were to count RNAs spots manually (as in their validation dataset), it would be only a matter of days. It is therefore perfectly doable in a short time.

In conclusion, in order to publish their manuscript, the authors should requantify their smFISH images and provide counts of cytoplasmic and nuclear HAC1 RNA spots, in WT and *isw1DELTA* cells, in Figure 4c.

REVIEWERS' COMMENTS

Reviewer #3 (Remarks to the Author):

In this second round of revision, the authors have addressed two of the three remaining points. However, my previous third point, dealing with the way the smFISH images are quantified to measure cytoplasmic and nuclear RNAs, is still not addressed adequately.

The validation of their method in Figure S5b is not a good validation because it compares cytoplasmic RNA on one side and nuclear/cytoplasmic ratio on the other. It is not the same thing, and this validation does not prove that the method counts correctly nuclear RNAs. Specifically, I believe that the out-of-focus signal coming from the many cytoplasmic RNAs could confound the measurements of nuclear RNAs. Moreover, the fact that the method was used before does not make it immune to flaws and artefacts.

The author claims that traditional RNA counting, done by counting the number of spots, is not accurate enough and they use this as a justification to use a method that is then much worse. I don't see the logic here. Again, direct measurement of nuclear and cytoplasmic RNAs is an essential point of the study and the authors should therefore employ the best possible method at hand and should not compromise their efforts.

The authors already have the smFISH images and even if they were to count RNAs spots manually (as in their validation dataset), it would be only a matter of days. It is therefore perfectly doable in a short time.

In conclusion, in order to publish their manuscript, the authors should requantify their smFISH images and provide counts of cytoplasmic and nuclear *HAC1* RNA spots, in WT and *isw1*DELTA cells, in Figure 4c.

We have evaluated, as requested by the reviewer, the number of individual *HAC1* transcripts in WT and *isw1* Δ cells, as now featured in Supplementary Figure 5c and detailed in the corresponding figure legend:

“The number of cytoplasmic *HAC1* transcripts was evaluated by manual counting of about 150 cells per strain. Note that all cells displayed a unique nuclear dot of variable intensity, corresponding to the transcription site, at which various amounts of nuclear-retained RNAs congregate. This method is therefore complementary to the global intensity quantification approach applied in Fig. 4c. The frequency distribution of cytoplasmic *HAC1* transcripts per bin (width 2) is represented for WT and *isw1* Δ . Two independent experiments were evaluated with similar results”.

This novel analysis and our previous quantification similarly establish that *Isw1* regulates the retention of *HAC1* mRNAs, as now mentioned in the text (line 194):

To further strengthen this observation, we quantified, the intensities of the *HAC1* mRNA signal in the nucleus and the cytoplasm of untreated cells, an approach that we validated on a manually quantified and previously published data set²⁶ (Supplementary Fig. 5b) as well as by evaluating the number of cytoplasmic *HAC1* transcripts through manual counting (Supplementary Fig. 5c).